# The ciliopathy protein CCDC66 controls mitotic progression and cytokinesis by promoting microtubule nucleation and organization

Umut Batman[1], Jovana Deretic[1], Elif Nur Firat-Karalar[1,2]*

1 Department of Molecular Biology and Genetics, Koç University, Istanbul, Turkey, 2 Koç University School of Medicine, Istanbul, Turkey

* ekaralar@ku.edu.tr

**Data Availability Statement:** All relevant data are within the paper and its Supporting Information files.

## Abstract

Precise spatiotemporal control of microtubule nucleation and organization is critical for faithful segregation of cytoplasmic and genetic material during cell division and signaling via the primary cilium in quiescent cells. Microtubule-associated proteins (MAPs) govern assembly, maintenance, and remodeling of diverse microtubule arrays. While a set of conserved MAPs are only active during cell division, an emerging group of MAPs acts as dual regulators in dividing and nondividing cells. Here, we elucidated the nonciliary functions and molecular mechanism of action of the ciliopathy-linked protein CCDC66, which we previously characterized as a regulator of ciliogenesis in quiescent cells. We showed that CCDC66 dynamically localizes to the centrosomes, the bipolar spindle, the spindle midzone, the central spindle, and the midbody in dividing cells and interacts with the core machinery of centrosome maturation and MAPs involved in cell division. Loss-of-function experiments revealed its functions during mitotic progression and cytokinesis. Specifically, CCDC66 depletion resulted in defective spindle assembly and orientation, kinetochore fiber stability, chromosome alignment in metaphase as well as central spindle and midbody assembly and organization in anaphase and cytokinesis. Notably, CCDC66 regulates mitotic microtubule nucleation via noncentrosomal and centrosomal pathways via recruitment of gamma-tubulin to the centrosomes and the spindle. Additionally, CCDC66 bundles microtubules in vitro and in cells by its C-terminal microtubule-binding domain. Phenotypic rescue experiments showed that the microtubule and centrosome-associated pools of CCDC66 individually or cooperatively mediate its mitotic and cytokinetic functions. Collectively, our findings identify CCDC66 as a multifaceted regulator of the nucleation and organization of the diverse mitotic and cytokinetic microtubule arrays and provide new insight into nonciliary defects that underlie ciliopathies.

**Funding:** This work was supported by European Research Council Starting Grant 679140 to ENF (https://erc.europa.eu/funding/starting-grants), European Molecular Biology Organization Installation Grant 3622 to ENF (https://www.embo.org), European Molecular Biology Organization Young Investigator Award 2020 to ENF (https://www.embo.org), The Scientific and Technological Research Council of Turkey BIDEB 120C148 grant to ENF (https://www.tubitak.gov.tr) and a Marie Sklodowska-Curie Fellowship 896644 to JD (https://marie-sklodowska-curie-actions.ec.europa.eu/actions/postdoctoral-fellowships). The funders had nor ole in study design, data collection and analysis, decision to publish, or preparation of the manuscript.

**Competing interests:** The authors have declared that no competing interests exist.

**Abbreviations:** AMR, actomyosin ring; CCDC66, coiled-coil domain-containing protein 66; CDK5RAP2, CDK5 regulatory subunit-associated protein 2; Cep55, centrosomal protein of 55 kDa; CEP152, centrosomal protein of 152 kDa; CEP192, centrosomal protein of 192 kDa; CEP290, centrosomal protein of 290 kDa; CPC, chromosomal passenger complex; CSPP1, centrosome and spindle pole-associated protein 1; EB1, end binding 1; GFP, green fluorescent protein; IFT88, intraflagellar transport protein 88; Kif23, kinesin-like protein KIF23; MKLP1, mitotic kinesin-like protein; K-fiber, kinetochore fiber; MAP, microtubule associated protein; MBP, maltose-binding protein; mNG, mNeonGreen protein; MT, microtubule; PCM, pericentriolar material; PLK1, polo-like kinase 1; PRC1, protein regulator of cytokinesis 1; STLC, S-trityl-L-cysteine; siRNA, small interfering RNA; TOGARAM1, TOG array regulator of axonemal microtubules protein 1; U-ExM, ultrastructure expansion microscopy.

# Introduction

Faithful segregation of genetic and cytoplasmic material during cell division is essential for growth and development of multicellular organisms. Deregulation of the molecular processes that regulate cell division leads to aneuploidy and chromosomal instability and thereby to the initiation and progression of various human cancers [1,2]. As such, mitosis and cytokinesis are highly regulated, multistep processes involving dynamic regulation and coordinated activity of multiple cellular structures and signaling pathways [3–5]. In particular, microtubule (MT) cytoskeleton undergoes a series of morphological changes to form diverse MT arrays such as the bipolar spindle, central spindle, and midbody. Precise spatiotemporal control of the assembly, maintenance, and dynamic remodeling of these MT arrays requires a diverse group of the MT-associated proteins (MAPs), which bind to MTs and regulate their dynamic properties, organization, and stability as well as their interactions with other proteins and cellular structures [6,7]. Thereby, MAPs play essential roles during numerous cell cycle processes including MT nucleation, formation, and organization of the mitotic spindle and central spindle, chromosome capture, alignment and segregation, and cleavage furrow formation and abscission [7–9]. Proteomic profiling of MT-based structures of dividing cells, functional screens, and loss-of-function studies have identified hundreds of MAPs as regulators of mitosis and cytokinesis [7,10]. However, key questions remain about their functions, mechanisms, and links to disease as well as how they cooperate with different cellular structures (i.e., centrosomes), protein complexes, and signaling pathways to modulate the parameters that ultimately define the size, shape, and dynamics of MT arrays.

In animal somatic cells, MT nucleation is initiated at the centrosomes, the preexisting spindle MTs, and the chromatin, with centrosomes being the major MT-organizing centers [9,11]. The mechanisms by which these distinct pathways work, their relative contributions to formation of diverse MT arrays in cells, and the extent of their crosstalk have been an area of active investigation. As cells enter mitosis, pericentriolar material (PCM) around centrioles expands in a process called centrosome maturation, which increases its MT-nucleation capacity [12,13]. Centrosome maturation is initiated by PLK1-dependent phosphorylation of pericentrin and CDK5RAP2, which promotes recruitment of additional PCM proteins including Cep152, Cep192, and gamma-tubulin [14–21]. Acentrosomal MT nucleation during mitosis is triggered at the chromosomes in RanGTP, Op8/Stathmin, and the chromosomal passenger complex (CPC)-dependent pathways, and at the spindle MTs in a HAUS/augmin complex-dependent way [9,22–25].

Organization of MTs into highly ordered mitotic and cytokinetic arrays play critical roles for cell division. For example, 20 to 40 kinetochore MTs in human cells form parallel bundles termed K-fibers, which run from spindle poles to kinetochores and are essential for chromosome alignment and segregation [26–29]. Bridging fibers, composed of antiparallel bundles of interpolar MTs, connect 2 sister K-fibers and push them apart to separate spindle poles [29,30]. Similarly, in anaphase, an antiparallel MT bundle forms the central spindle/midzone that pushes the spindle poles to opposite side of the cell and directs the localization of ingression furrow important for the division of cytoplasm [8]. Antiparallel MT bundles at the spindle midzone are cross-linked by the evolutionarily conserved Protein Translocator of Cytokinesis 1 (PRC1)/Ase1/MAP65 family [31–34]. Central spindle shortens to form the midbody during cytokinesis, which will direct the membrane abscission site [35]. Moreover, astral MTs, which emanate from the centrosomes, interact with the cell cortex to position the spindle within a cell and determine the initial cleavage plane through communication with the equatorial cortex [36]. Although multiple MAPs involved in the formation and stabilization of the distinct

spindle bundles have been identified, the full extent of MAPs involved in MT cross-linking and stability as well as their mechanism of action have yet to be determined in future studies.

The drastic remodeling of the MT network during cell division requires precise regulation of when and where MAPs are activated. A subset of MAPs is active during mitosis but inactive during interphase. Such regulation is achieved by modulation of their affinity to MTs, regulation of their cellular abundance, and localization and posttranslational modifications [37]. Importantly, there is also a group of MAPs with dual functions in dividing and nondividing cells [38–40]. For example, End Binding 1 (EB1) regulates MT plus-end dynamic and targets other MAPs to the plus ends both in interphase and mitosis [41–44]. Recently, a critical regulator of primary cilium assembly in nondividing cells, intraflagellar transport protein 88 (IFT88), has been described for its functions during mitotic spindle orientation and central spindle organization [45,46]. Importantly, the discovery of nonciliary functions of IFT88 unraveled that its mutations might contribute to polycystic kidneys with both impaired ciliary function and aberrant cell division [45,46]. Despite the progress made in the characterization of MAPs with dual roles in cycling and noncycling cells, questions remain about their functions, mechanisms, and modes of regulation in different stages of the cell cycle.

We previously characterized coiled coil protein 66 (CCDC66) as a MAP and a regulator of primary cilium formation and composition in quiescent cells [47,48]. It was originally described as a gene mutated in retinal degeneration and later characterized for its retinal and olfactory functions using CCDC66$^{-/-}$ mouse [49–52]. Recently, CCDC66 was identified as part of the Joubert syndrome interaction network consisting of other MAPs such as CSPP1, TOGARAM1, and CEP290 [53]. Consistent with its link to ciliopathies, we and others previously showed that retinal degeneration mutations disrupt its ciliary functions and interactions [48,51]. In addition to its ciliary functions, following lines of evidence suggest that CCDC66 might function as a regulator of cell division: CCDC66 mRNA was identified in the MT-interacting transcriptome of *Xenopus tropicalis*, indicative of its functions during MT-based cellular processes [54]. Additionally, CCDC66 localized to centrosomes and MTs in dividing cells and its depletion led to disorganized poles in mitotic cells [48,54]. Finally, CCDC66 proximity interactome generated from asynchronous cells revealed interactions with regulators of cell division [48,55,56]. However, the full extent of CCDC66 functions during different stages of cell division and the underlying molecular mechanisms are not known. Addressing these key unknowns will uncover the relationship of CCDC66 with other components of the mitotic and cytokinetic machinery and also provide insight into whether and if so, how its nonciliary functions contribute to its disease mechanisms.

In this study, we examined the localization, interactions, functions, and mechanisms of CCDC66 during cell division. We showed that CCDC66 is required for recruitment of core machinery of centrosome maturation to the centrosomes and acts as a bundling protein in vitro and in cells. Its association with centrosomes and MTs is required for spindle assembly and organization, K-fiber and midbody integrity, chromosome alignment, and cytokinesis. Our findings unravel nonciliary functions for CCDC66 during cell division and provide insight into the integrated activity of centrosomes and MAPs during spatiotemporal regulation of MT nucleation and organization in mitosis and cytokinesis.

## Results

### CCDC66 localizes to centrosomes and MTs during mitosis and cytokinesis

Based on its previously reported localization and interactions, we hypothesize that CCDC66 plays important roles during mitosis and cytokinesis [48,54,55]. To test this, we first examined localization of endogenous and mNeonGreen-CCDC66 fusion proteins at different cell cycle

stages in mammalian cell lines. Antibody against endogenous CCDC66 revealed its localization to the centrosome throughout the cell cycle (Figs 1A and S1A). In dividing cells, CCDC66 also localized to multiple MT-based structures including the spindle MTs in prometaphase and metaphase, the central spindle in anaphase, and the midbody in cytokinesis in human osteosarcoma (U2OS) cells (Figs 1A and S1A). To examine the dynamic localization of CCDC66 during cell cycle, we generated cell lines that stably express mNeonGreen (mNG)-CCDC66 using lentiviral transduction. mNG protein was chosen over green fluorescent protein (GFP) as the fluorescent tag due to its higher fluorescent intensity and stability [57]. Stable expression of the fusion protein in U2OS and RPE1 cells was validated by immunoblotting using mNG antibody and immunofluorescence using anti-CCDC66 antibody (S1B–S1D Fig). In fixed U2OS and RPE1 stable cells, mNG-CCDC66 localized to the centrosome and centriolar satellites in interphase cells (Figs 1B and S1E) and to the centrosomes and MT-based structures of mitosis and cytokinesis in dividing cells (Figs 1B and S1E). Although CCDC66 localization to the astral MTs and midzone was apparent in cells stably expressing mNG-CCDC66, it was very weak in cells stained for endogenous CCDC66 (Figs 1A and S1A). This might be due to the high cytoplasmic and punctate background associated with CCDC66 antibody staining and/or the relatively lower abundance of CCDC66 at the astral microtubules and spindle midzone. In agreement with its localization in fixed cells, time-lapse imaging of mNG-CCDC66 cells stained with SIR-tubulin showed that CCDC66 dynamically localized to the centrosome, centriolar satellites, and MT-based structures in different cell states (S1F and S1G Fig and S1 and S2 Movies).

Previously, we showed that CCDC66 and its N-terminal 1–570 and C-terminal 570–948 amino acid residue fragments localize to MTs in cells [48]. Given that C-terminal fragment binds to MTs directly, we tested whether this C-terminal fragment recapitulates the localization of full-length CCDC66 during cell division. Like mNG-CCDC66, mNG-CCDC66 (570–948) localized to the centrosomes, bipolar spindle, and central spindle during mitosis and midbody during cytokinesis in U2OS cells (Fig 1C). When overexpressed, mNG-CCDC66 (570–948) formed cytoplasmic aggregates that recruited gamma-tubulin, suggesting a putative interaction between them (Fig 1D). To investigate the functional significance of this recruitment during MT nucleation, we performed MT regrowth experiments and found that the cytoplasmic aggregates nucleated MTs 10 min after nocodazole washout (Fig 1D). Together, these data indicate that the C-terminal 379 residues of CCDC66 are sufficient for its cellular localization to the centrosome and MTs in dividing cells.

## CCDC66 co-localizes and interacts with critical regulators of mitosis and cytokinesis

We defined the high-resolution localization of CCDC66 relative to the markers of the centrosomes, bipolar spindle, central spindle, and midbody. Specifically, we co-stained U2OS:: mNG-CCDC66 cells with antibodies against PCM proteins gamma-tubulin, CEP192, CEP152, CDK5RAP2 (centrosomes), the MT-cross-linking protein PRC1 (central spindle and midbody MTs), the centrosome protein CEP55 (midbody core), the kinesin motor protein Kif23/MKLP1, and phosphorylated Aurora A/B/C (midbody flanks) [8,35,58]. CCDC66 localized to the centrosomes throughout mitosis, as shown by its co-localization with gamma-tubulin, CDK5RAP2, CEP192, and CEP152 (Fig 2A and 2B). Notably, the centrosome-associated pool of CCDC66 was maintained in nocodazole-treated cells, confirming that this pool binds to the centrosomes independent of MTs (Fig 2C). To determine the precise cytokinetic localization of CCDC66, we performed plot profile analysis along the midbody MT bundles for CCDC66 and known midbody markers (Fig 2D) [35]. In cytokinesis, CCDC66 localized to 2 closely

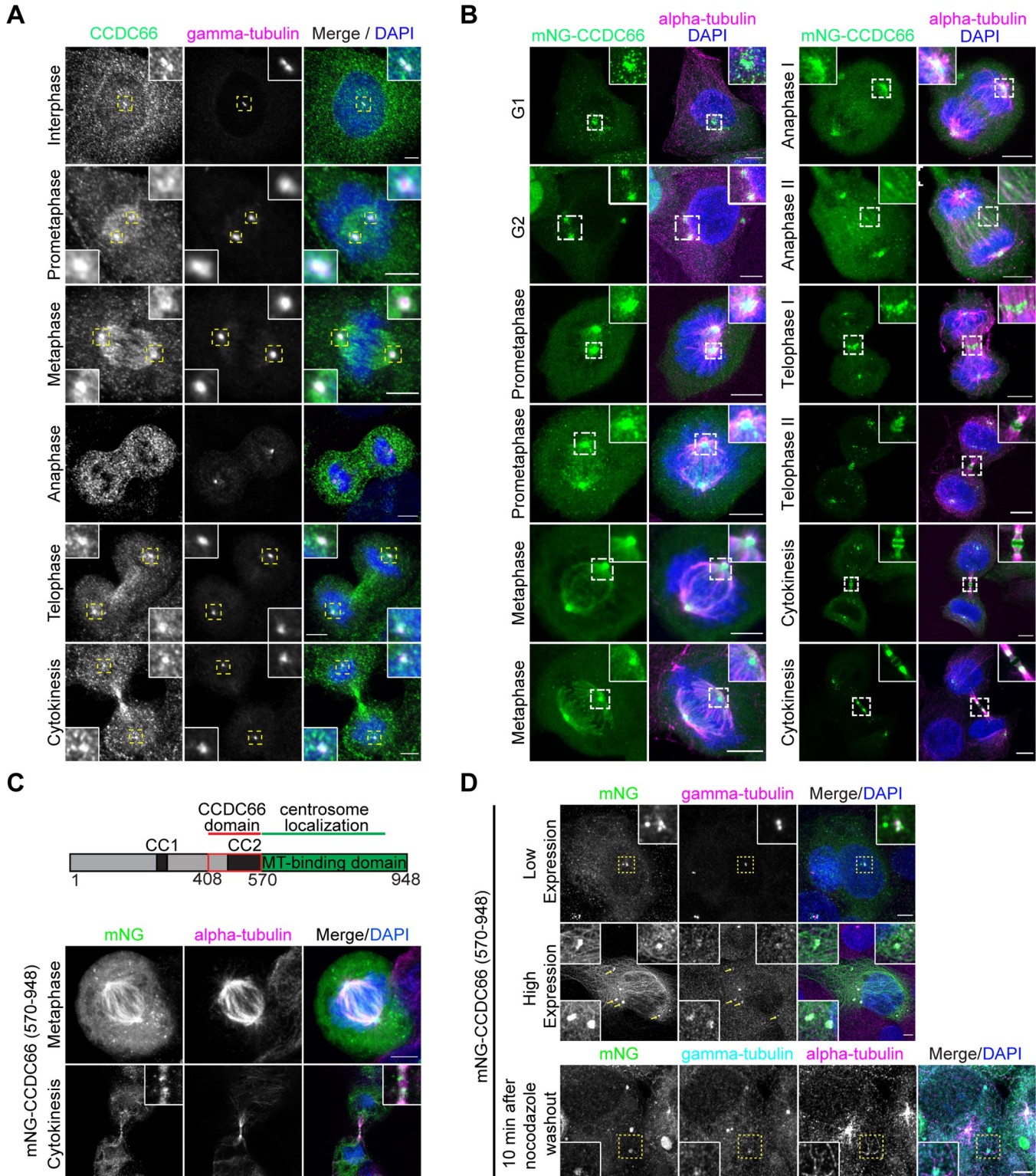

**Fig 1. CCDC66 localizes to centrosomes and microtubule-based structures during cell division.** (A) Localization of CCDC66 at different stages of the cell cycle. U2OS were fixed with methanol followed by acetone and stained for CCDC66, gamma-tubulin, and DAPI. Scale bar: 5 μm, insets show 4× magnifications of the boxed regions. (B) Localization of mNeonGreen-CCDC66 at different stages of the cell cycle. U2OS cells stably expressing mNeonGreen-CCDC66 fusion (U2OS::mNG-CCDC66) were fixed with 4% PFA and stained for alpha-tubulin and DAPI. Scale bar: 5 μm, insets show 4× magnifications of the boxed regions. (C) Schematic representation of FL CCDC66 domain organization. CC1 and CC2 indicate coiled-coil domains. The C-terminal region was

previously described as a microtubule-binding region [48]. Still confocal images show U2OS cells transfected with mNeonGreen-CCDC66 C-terminal construct (mNG-CCDC66$^{570-948}$). Approximately 24 h post-transfection, cells were fixed with 4% PFA and stained for alpha-tubulin and DAPI. mNG-CCDC66$^{570-948}$ localizes to centrosomes and spindle microtubules during metaphase and to midbody during cytokinesis. Scale bar: 5 μm. (D) mNG-CCDC66$^{570-948}$ sequesters gamma-tubulin to cytoplasmic aggregates. U2OS cells were transfected with mNG-CCDC66$^{570-948}$, fixed with 4% PFA, and stained for gamma-tubulin and DAPI. Top panel shows a lower expressing cell in which mNG-CCDC66$^{570-948}$ is restricted mostly to centrosomes. Middle panel shows a high-expressing cell with multiple cytoplasmic aggregates of mNG-CCDC66$^{570-948}$ that co-localize with gamma-tubulin. Bottom panel shows mNG-CCDC66$^{570-948}$ transfected cells that are treated with 5 μg/ml nocodazole for 1 h at 37°C and incubated for 10 min after nocodazole washout. Scale bar: 5 μm, insets show 4× magnifications of the boxed regions. CC, coiled-coil domain; CCDC66, coiled-coil domain-containing protein 66; DAPI, 4′,6-diamidino-2-phenylindole; FL, full length; PFA, paraformaldehyde.

spaced bands at the midbody. Plot profile analysis revealed its co-localization with PRC1 at the dark zone (Fig 2D), which is defined as the narrow region on the MT bundle in the center of the midbody [35]. mNG-CCDC66 also co-localized with PRC1 at the spindle midzone in anaphase and telophase cells (S2A Fig). In contrast, CCDC66 did not co-localize with CEP55 and MKLP1 at the midbody, which are visualized as rings representing the bulge at the center of the midbody, and phospho-Aurora (Fig 2D), which marks the broader bands on MTs outside the dark zone [35]. Of note, we observed accumulation of CCDC66 outside the midbody in cells at later stages of cytokinesis, which is also evident in the dynamic behavior of mNG-CCDC66 during cell division (Fig 2D and S1 and S2 Movies). These CCDC66-positive structures did not co-localize with the known midbody markers and were not detected in cells stained for endogenous CCDC66 (Fig 1A).

To generate hypothesis regarding the functions and mechanisms of CCDC66 during cell division, we compiled a list of CCDC66 interactors from high throughput proximity-mapping studies and performed Gene Ontology (GO)-enrichment analysis of its interactors (S2B and S2C Fig) [48,55,56]. In addition to proteins involved in primary cilium assembly and function, CCDC66 proximity interactors were enriched for biological processes related to cell division including spindle assembly and organization, chromosome segregation, metaphase plate congression, and cytokinesis (S2B Fig). Consistently, cellular compartments involved in these processes such as the HAUS complex, mitotic spindle, spindle pole/centrosomes, and kinetochore were among the enriched GO categories (S2C Fig). This analysis supports nonciliary functions for CCDC66 during cell division.

To define the physical mitotic and cytokinetic interactions of CCDC66, we performed Flag-based co-immunoprecipitation experiments with PCM proteins and mitotic MAPs that co-localize with CCDC66 (Fig 2A–2D) or were described as critical regulators of mitosis and cytokinesis. As a positive control for interaction experiments, we used the centriolar satellite marker protein PCM1, which we previously reported as an interactor for CCDC66 [48]. First, we assayed whether CCDC66 interacts with critical regulators of centrosome maturation. CCDC66 co-pelleted with myc-BirA* fusions of CDK5RAP2, Cep192, Cep152, and gamma-tubulin (Fig 2E). As negative controls, FLAG-miniTurbo did not co-pellet with myc-BirA* fusions of these positive interactions, and CCDC66 did not co-pellet with the MT plus-end-tracking protein EB1 (Fig 2E). Next, we tested interactions of CCDC66 with regulators of cytokinesis [8]. Flag-miniTurbo-CCDC66 interacted with the GFP-PRC1, but not GFP-CEP55 and the negative control GFP (Fig 2F). Together with its localization profile during cell division, the new CCDC66 interactors we identified suggest potential mitotic and cytokinetic functions of CCDC66 via regulating centrosome maturation, MT nucleation, and/or organization.

## CCDC66 is required for mitotic progression and cytokinesis

To elucidate CCDC66 functions during mitotic and cytokinetic progression, we performed loss-of-function experiments using a siRNA validated in depletion and rescue experiments

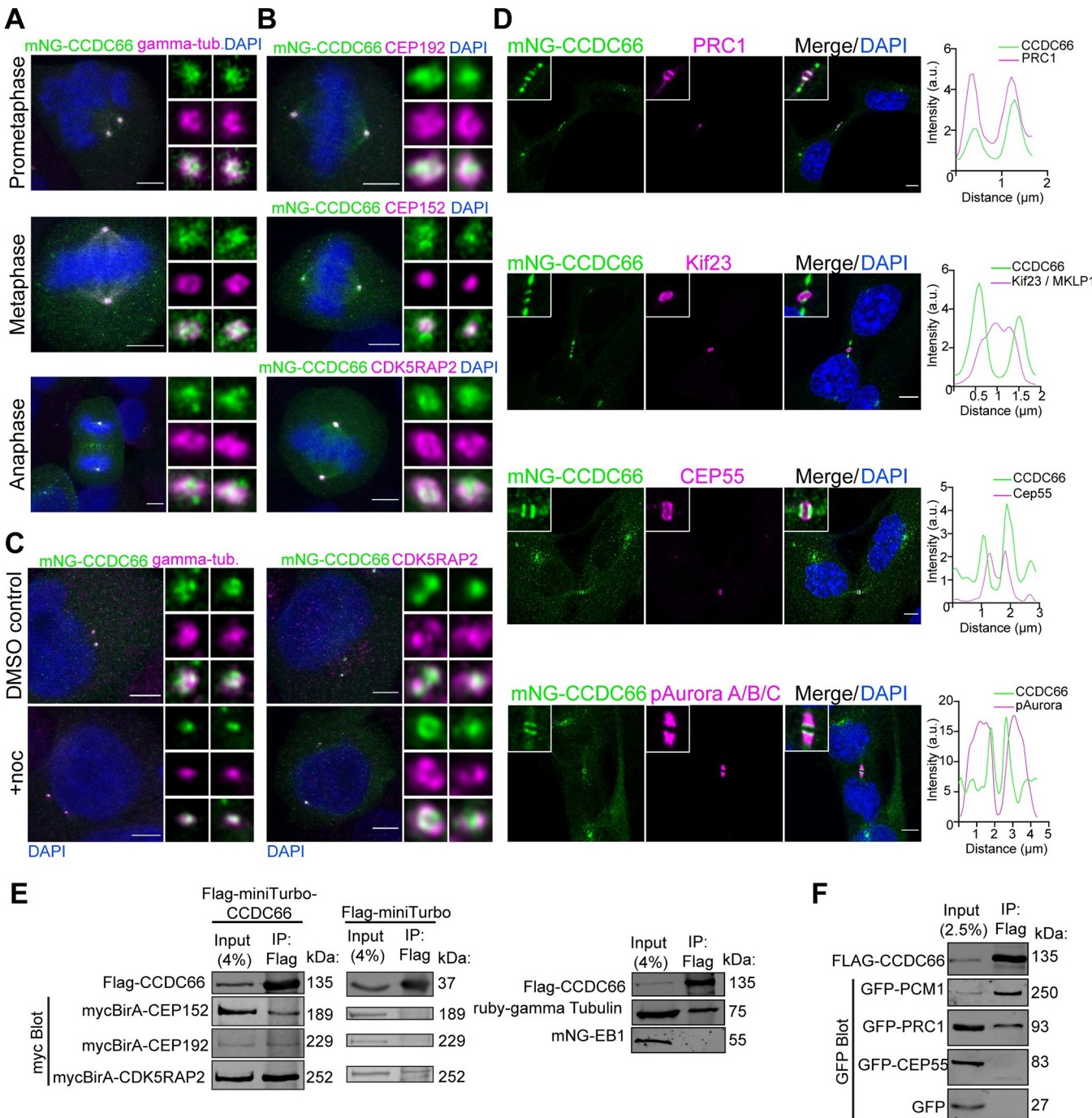

**Fig 2. CCDC66 co-localizes and interacts with PCM proteins and spindle/midbody MAPs.** (A) mNG-CCDC66 localizes to the centrosome throughout the cell cycle. U2OS::mNG-CCDC66 cells were fixed with 4% PFA and stained for gamma-tubulin and DAPI. Still deconvolved confocal images represent centrosomal co-localization of mNG-CCDC66 with gamma-tubulin during different stages of cell division. Scale bar: 5 μm, insets show 4× magnifications of the centrosomes. (B) Localization of mNG-CCDC66 in U2OS cells relative to PCM proteins. U2OS::mNG-CCDC66 cells were fixed with 4% PFA and stained for CEP192, CEP152, or CDK5RAP2 and DAPI. Deconvolved confocal images represent their relative localization at the centrosomes during metaphase. Scale bar: 5 μm, insets show 4× magnifications of the centrosomes. (C) Effect of microtubule depolymerization on CCDC66 localization at the centrosome. U2OS::mNG-CCDC66 cells were treated with 0.1% DMSO or 5 μg/ml nocodazole for 1 h. Cells were then fixed with 4% PFA and stained for gamma-tubulin or CDK5RAP2 and DAPI. Scale bar: 5 μm, insets show 4× magnifications of the centrosomes. (D) Localization of mNG-CCDC66 in RPE1 cells relative to midbody proteins. RPE1::mNG-CCDC66 cells were fixed with methanol and stained for mNeonGreen and PRC1, Cep55, Kif23 (MKLP1), or pAurora A/B/C and DAPI. Graphs show the plot profiles to assess co-localization with the indicated marker. Using ImageJ, a straight line was drawn on the midbody and intensity along the distance was plotted on Graphpad Prism. (E) Co-immunoprecipitation of Flag-miniTurbo-CCDC66 or Flag-miniTurbo with PCM proteins

from HEK293T cells. Flag-miniTurbo-CCDC66 was co-transfected with myc-BirA-Cep192, myc-BirA-Cep152, and myc-BirA-CDK5RAP2 in HEK293T cells. Flag-miniTurbo-CCDC66 and Flag-miniTurbo were precipitated with Flag beads and input and eluates (IP) were blotted with Flag and myc antibodies to assess the interaction. HEK293T cells were also co-transfected with Flag-miniTurbo-CCDC66 and Ruby-Gamma-tubulin-T2A-mNG-EB1 fusion construct. Flag-miniTurbo-CCDC66 was precipitated with Flag beads and Input and eluates (IP) were blotted with Flag, EB1, and gamma-tubulin antibodies. (F) Co-immunoprecipitation of Flag-CCDC66 with midbody proteins from HEK293T cells. Flag-CCDC66 was co-transfected with GFP, GFP-Cep55, or GFP-PRC1 in HEK293T cells. Flag-CCDC66 was precipitated with Flag beads and Input and eluates (IP) were blotted with Flag and GFP antibodies to assess the interaction. The data underlying the graphs showing the plot profiles in the figure can be found in S1 Data. CCDC66, coiled-coil domain-containing protein 66; CDK5RAP2, CDK5 regulatory subunit-associated protein 2; Cep55, centrosomal protein of 55 kDa; CEP192, centrosomal protein of 192 kDa; CEP152, centrosomal protein of 152 kDa; DAPI, 4′,6-diamidino-2-phenylindole; DMSO, dimethyl sulfoxide; GFP, green fluorescent protein; Kif23, kinesin-like protein KIF23 (MKLP1, mitotic kinesin-like protein 1); pAurora A/B/C, phospho-AuroraA/B/C; PCM, pericentriolar material; PFA, paraformaldehyde; PRC1; protein regulator of cytokinesis 1.

[48]. Given that CCDC66 localization and dynamics in dividing cells were similar in both RPE1 and U2OS cells (Figs 1 and S1), we chose U2OS cells for further characterization as a p53-responsive transformed cell. Immunoblotting and immunofluorescence analysis of U2OS cells 48 h after transfection with control and CCDC66 siRNAs confirmed efficient depletion (S3A and S3B Fig). To investigate the role of CCDC66 in the dynamic events of the cell cycle, we performed time lapse confocal imaging of control and CCDC66-depleted U2OS cells that stably express the chromosome marker mCherry-H2B and determined the fate of the dividing cells (Fig 3A). We plotted the fate of individual control and CCDC66-depleted cells as vertical bars in S2C Fig, where the height of the bar represents the mitotic time and the color of the bars represent the different fates (gray: successful division, pink: mitotic arrest, cyan: apoptosis). The mitotic time, which was defined as the time from nuclear envelope breakdown to anaphase, increased about 1.4-fold in CCDC66-depleted cells relative to control cells (siCCDC66: 39.2 ± 8.6 min, siControl: 27.1 ± 6.6 min) (Fig 3B and S3 and S4 Movies). The mitotic cells that did not complete mitosis during 12 h of live imaging had 2 different fates. They either exhibited prometaphase arrest, with some cells reaching chromosome alignment followed by metaphase plate regression or underwent apoptosis, which was assessed by membrane blebbing and DNA fragmentation (Fig 3A, 3C, and 3D and S5–S7 Movies). As compared to 8.8 ± 3.6% of control cells, 23.3 ± 3.7% percent of CCDC66-depleted cells died after prolonged mitosis (Fig 3D). We also quantified the mitotic index by scoring the percentage of cells positive for the mitotic marker phospho-H3 and found that it was not altered upon CCDC66 loss (S3D Fig). This might be due to increased apoptosis in U2OS cells as a response to the delayed mitosis followed by mitotic failure.

In addition to mitotic defects, we observed that CCDC66 depletion interfered with progression of cytokinesis. Time lapse imaging of dividing CCDC66-depleted cells revealed 2 different events that resulted in binucleated cells (Fig 3A and S8 and S9 Movies). First phenotype was the regression of the cleavage furrow after chromosome segregation. Second phenotype was failure in forming the cleavage furrow and initiating cytokinesis (Fig 3A and S8 and S9 Movies). In a complementary approach, we quantified the percentage of binucleated cells in a time course manner by fixing cells at different time points after siRNA transfection. CCDC66 depletion caused an increase in the percentage of binucleated cells in a time course manner (Fig 3E). These results identify CCDC66 as a regulator of cytokinesis.

To further define its functions during mitotic progression, we examined whether CCDC66 depletion leads to the accumulation of cells at a particular stage of mitosis by scoring cells based on their chromosome positioning and spindle organization using DNA and MT staining (Fig 3F) [59]. CCDC66 depletion led to a higher fraction of cells in prometaphase, suggesting the presence of chromosome and/or spindle-related defects (Fig 3F and 3G). In agreement, we noted that the MT arrays of the bipolar spindle, central spindle, and midbody of CCDC66-depleted cells were disorganized (Fig 3F). To quantify chromosome alignment defects, we

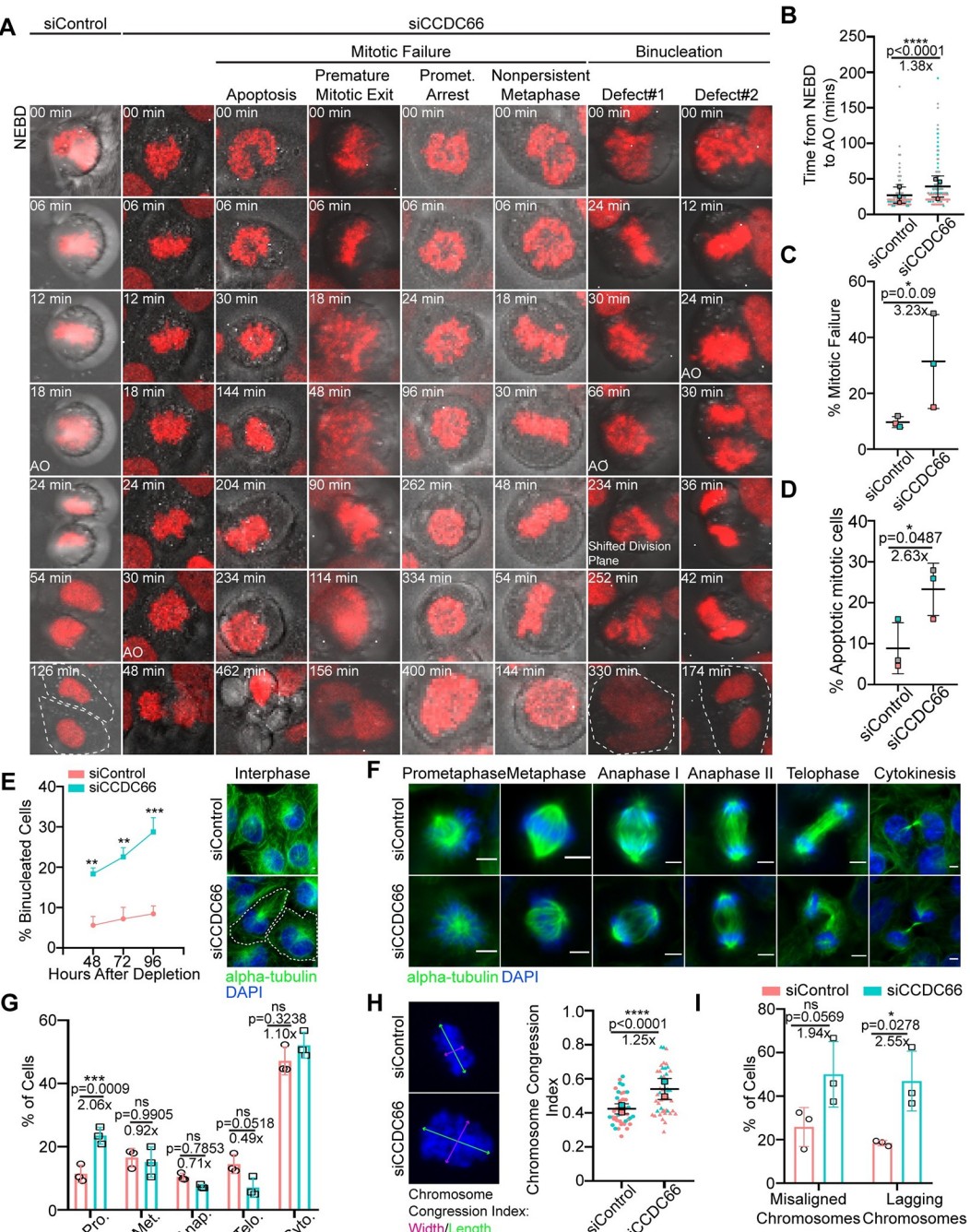

**Fig 3. CCDC66 is required for faithful mitotic progression and cytokinesis.** (A) Effects of CCDC66 depletion on mitotic progression. U2OS::mcherry-H2B cells were grown on 2-well Lab-Tek plates and transfected with nontargeting (siControl) or CCDC66 targeting (siCCDC66) siRNA. After 48 h of transfection, cells were imaged with confocal microscopy with 20× objective. Images are acquired every 6 min for 12 h. Representative still images from live imaging are shown for different phenotypic categories for siCCDC66 include apoptosis, premature mitotic exit, prometaphase arrest, and nonpersistent metaphase. (B) Quantification of mitotic time from (A). Mitotic time was quantified as the time interval from NEBD to AO. Data represent the mean ± SEM of 3 independent experiments and is plotted using super-plot. (C) Quantification of percent mitotic failure from A. Mitotic failure refers to the cells that entered mitosis but could not proceed through anaphase because of: premature mitotic exit, prometaphase arrest, and nonpersistent metaphase. Data represent the mean ± SEM of 3 independent experiments and is plotted using super-plot. (D) Quantification of percent mitotic cell death from (A). Mitotic cell death represents mitotic cells that underwent apoptosis during the time of imaging. Data represent the mean ± SEM of 3 independent experiments and is plotted using super-plot. (E) CCDC66 depletion increases binucleation. Cells were transfected with either siControl or siCCDC66; fixed with methanol 48, 72, and 96 h post-transfection; and stained for alpha-

tubulin and DAPI. Binucleation was categorized based on DNA and alpha-tubulin staining. Scale bar: 5 μm. (F) Spindle and chromosome phenotypes of CCDC66-depleted cells. Representative images of different stages of cell division for siControl and siCCDC66 transfected U2OS cells are shown. Cells were transfected with either siControl or siCCDC66, fixed with methanol 48 h post-transfection, and stained for alpha-tubulin and DAPI. Mitotic cell stages were categorized based on DNA staining. Scale bar: 5 μm. (G) Quantification of (E). Data represent mean ± SEM of 3 independent experiments. $n > 1,000$ for all experiments. Mean prometaphase percentage is 11.42 for siControl and mean prometaphase percentage is 23.50 for siCCDC66. (***$p < 0.001$, ns: not significant). (H) CCDC66 depletion increases chromosome width. U2OS cells were transfected with siRNA, fixed with methanol, and stained for DAPI. Chromosome congression index is measured by dividing the length of the metaphase plate by its width. Data represent the mean ± SEM of 2 independent experiments. (****$p < 0.0001$). Scale bar: 5 μm. (I) Quantification of chromosome alignment and segregation defects from (A). The quantification shows the number of metaphase cells that have misaligned chromosomes and anaphase and telophase cells that have lagging chromosomes. Data represent the mean ± SEM of 3 independent experiments. (ns: not significant). The data underlying the graphs shown in the figure can be found in S1 Data. AO, anaphase onset; CCDC66, coiled-coil domain-containing protein 66; DAPI, 4′,6-diamidino-2-phenylindole; NEBD, nuclear envelope breakdown; SEM, standard error of mean; siRNA, small interfering RNA.

measured the ratio of the width to the height of the chromosomal mass, which was previously described as the chromosome congression index (Fig 3H) [60,61]. CCDC66-depleted cells had a higher chromosome congression index than control cells, which indicates defective chromosome alignment at the metaphase plate (Fig 3H). Moreover, the percentage of cells with lagging chromosomes, but not with misaligned chromosomes, increased in CCDC66-depleted cells relative to control cells (Fig 3I). Taken together, our findings demonstrate that CCDC66 regulates mitotic progression and cytokinesis in part via ensuring proper chromosome alignment and spindle assembly.

## CCDC66 is required for spindle assembly and orientation

The localization of CCDC66 to spindle microtubules suggests that mitotic progression defects associated with its depletion might be a consequence of defective spindle assembly and orientation. To investigate this, we analyzed various spindle properties in control and CCDC66-depleted cells. First, we measured the angle between the spindle axis and the substratum, which revealed an increase from 7.7 ± 0.3° in control cells to 13.1 ± 1.5° in CCDC66-depleted cells (Fig 4A). By taking into account the changes into spindle angle, we quantified spindle length and found that it was not altered upon CCDC66 depletion (Fig 4A). Likewise, centrosome width at metaphase was comparable between control and CCDC66-depleted cells (Fig 4A). The essential role of astral MTs in spindle positioning and orientation led us to investigate CCDC66 functions during astral MT assembly and stability [62]. To this end, we quantified the astral MT fluorescence intensity and length in control and CCDC66-depleted cells and found that they were both reduced upon CCDC66 loss (Fig 4B and 4C). Likewise, the tubulin fluorescence intensity at the spindle in metaphase cells decreased about 0.6-fold in CCDC66-depleted cells relative to control cells (Fig 4B and 4C). Immunoblotting of lysates prepared from control and CCDC66 siRNA-transfected cells showed that the intensity changes in spindle MTs were not due to altered cellular abundance of alpha-tubulin (S4A Fig). Together, these results show that CCDC66 is required for spindle MT assembly and stability.

Proper formation and organization of K-fibers is essential for chromosome alignment and segregation, suggesting that chromosome-related defects in CCDC66-depleted cells could be due to impaired K-fibers [29,63]. To determine whether CCDC66 is specifically required for the K-fiber stability, we performed cold stability assay in control and CCDC66-depleted cells. In this assay, K-fibers are visualized by selective depolymerization of less stable interpolar and astral MTs by cold treatment of cells for 10 min at 4°C [64,65]. The tubulin fluorescence intensity of cold-stable K-fibers was reduced in CCDC66-depleted cells relative to control cells (Fig 4D), which identify CCDC66 as a regulator of K-fiber stability. Notably, mNG-CCDC66 and

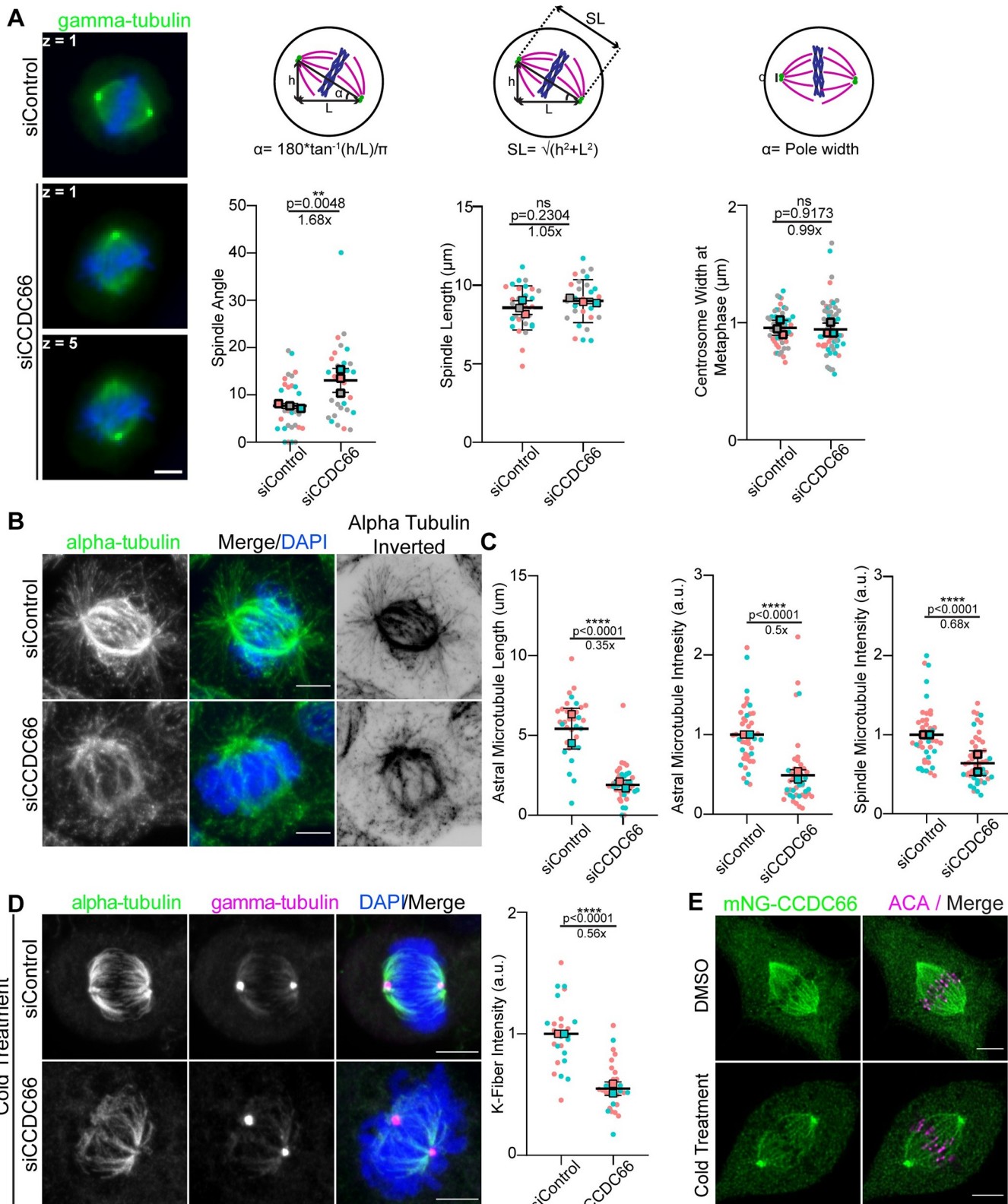

**Fig 4. CCDC66 regulates spindle organization and orientation.** (A) Effects of CCDC66 depletion on spindle angle, length, and pole width. U2OS cells were transfected with control and CCDC66 siRNA, fixed with methanol and stained for gamma-tubulin; z on still images indicates the stack where the centrosome is found. Spindle angle is calculated by the formula $\alpha = 180 \ast \tan^{-1}(h/L)/\pi$ where h represents the stack difference between 2 centrosomes, L represents the distance

between centrosomes. SL is calculated by the formula $SL = \sqrt{(h2+L2)}$ where h represents the stack difference between 2 centrosomes, L represents the distance between centrosomes when projected onto 1 z plane. Centrosome width at metaphase is calculated by measuring the length of the PCM of the centrosome. Data represent the mean ± SEM of 3 independent experiments. (** $p < 0.01$, ns: not significant). Scale bar: 5 μm. (B) Spindle microtubule density and astral microtubule length is reduced in CCDC66-depleted cells. U2OS cells were transfected with siRNA, then fixed with methanol, and stained for alpha-tubulin and DAPI. Representative images are shown. Inverted image is shown to emphasize astral microtubules better. Scale bar: 5 μm. (C) Quantification of (B). Astral microtubule and spindle microtubule intensity were measured on ImageJ by taking several points on the spindle to measure the intensity and subtracting the background mean intensity. Data represent the mean ± SEM of 3 independent experiments. (**** $p < 0.0001$). (D) CCDC66 depletion reduces K-fiber intensity. Cells were transfected with siRNA then 48 h after transfection, cells were incubated in ice for 10 min. Cells were fixed with methanol and stained for alpha-tubulin, gamma-tubulin, and DAPI. K-fiber intensity was measured as described for (C). Data represent the mean ± SEM of 2 independent experiments. (**** $p < 0.0001$). Scale bar: 5 μm. (E) CCDC66 localizes to K-fibers. RPE1::mNG-CCDC66 stable cell line was incubated in ice for 10 min and fixed with MeOH then stained for mNG and ACA. Images represent a single stack and were captured with the same camera settings from the same coverslip. Scale bar: 5 μm. The data underlying the graphs shown in the figure can be found in S1 Data. ACA, anticentromeric antibody; CCDC66, coiled-coil domain-containing protein 66; DAPI, 4′,6-diamidino-2-phenylindole; K-fiber, kinetochore-fiber; L, length; SEM, standard error of mean; siRNA, small interfering RNA; SL, spindle length.

endogenous CCDC66 still localized to the spindle microtubules in cold-treated cells, which confirms its localization to K-fibers (Figs 4E and S4C). Collectively, our findings indicate that CCDC66 regulates spindle assembly and orientation to ensure proper mitotic progression.

## CCDC66 is required for the assembly and organization of the central spindle and cleavage furrow

CCDC66 localizes to the central spindle, intercellular bridge/midbody, and K-fibers, which are highly organized and stable MT bundles. Moreover, it interacts and co-localizes with PRC1, a well-characterized regulator of MT bundling during cell division [33]. These lines of data suggest that CCDC66 might regulate assembly and organization of MTs at the central spindle and cleavage furrow. To test this, we examined how loss of CCDC66 affects assembly and organization of these MT arrays in dividing cells.

Quantification of the MT fluorescence intensity along the pole-to-pole axis in anaphase cells showed that MT intensity was reduced at the central spindle in CCDC66-depleted cells (Fig 5A and 5B). This result is analogous to the effects of CCDC66 depletion on spindle MT intensity of metaphase cells. Strikingly, CCDC66 depletion also severely disrupted the organization of central spindle MTs (Fig 5A and 5C). Majority of control cells (80.68%) had the typical central spindle organization characterized by 2 dense arrays of tightly packed MTs separated by a thin line at the cell center (Fig 5A) [66]. In contrast, a significantly higher fraction of CCDC66-depleted cells exhibited highly disorganized central spindles characterized by reduced and poorly aligned MTs (siControl: 20.32%, siCCDC66: 57.04%) (Fig 5A and 5C). The width of the midzone remained unaltered upon CCDC66 depletion (S5A Fig). In addition to the central spindle, CCDC66 loss compromised the geometry of the cleavage furrow and resulted in highly asymmetric cleavage furrow ingression (Fig 5D). About 63.8% of CCDC66-depleted cells exhibited this defect relative to 23.8% in control cells (Fig 5E). Taken together, these results demonstrate that CCDC66 is involved in the assembly and organization of central spindle and intercellular bridge MTs.

Defects in central spindle and intercellular bridge/midbody, as well as K-fibers could be attributed to MT bundling activity of CCDC66. To test this, we used in vitro experiments to investigate the possible MT cross-linking activity of CCDC66. We expressed MBP-mNG-CCDC66 in insect cells and purified it using nickel beads (S5C and S5D Fig). As a control, we purified MBP-mNG in bacteria (S5F Fig). Using in vitro MT sedimentation assay, we showed that MBP-mNG-CCDC66 directly binds to MTs (S5E Fig). After validation of MT affinity, we performed in vitro MT bundling assays. Incubation of MBP-mNG-CCDC66, but not MBP-mNG, with taxol-stabilized rhodamine-labeled MTs resulted in the formation of MT bundles in vitro (Fig 5F). Notably, these bundles co-localized with CCDC66, confirming its

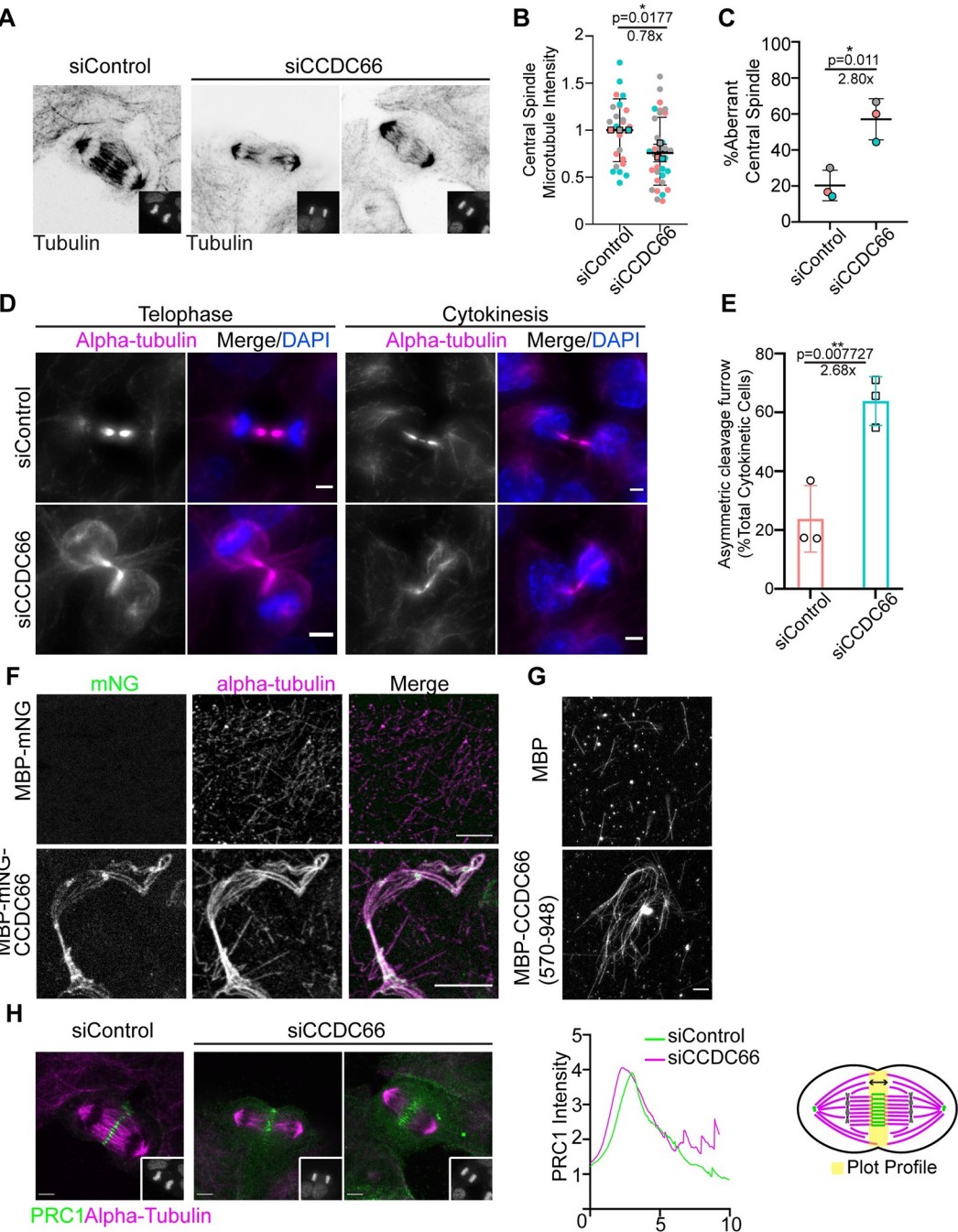

**Fig 5. CCDC66 bundles microtubules and is required for assembly and organization of the central spindle and cleavage furrow.** (A) Effect of CCDC66 depletion on central spindle assembly and organization. U2OS cells were transfected with control or CCDC66 siRNA, fixed with methanol followed 48 h post-transfection, and stained for alpha-tubulin and DAPI. Representative images show cells at late anaphase, as indicated by the DNA staining in the inset. (B) Quantification of (A). Graph represents the microtubule density at central spindle. Central spindle microtubule intensity was measured on ImageJ by taking several points on the spindle to measure the intensity and subtracting the background mean intensity. Data represent the mean ± SEM of 3 independent experiments. (*$p < 0.05$). (C) Quantification of (A). Graph represents the percentage of aberrant central spindle with mean ± SEM of 2 independent experiments. (*$p < 0.05$). Aberrant central spindle is characterized by reduced and poorly aligned microtubules. (D) CCDC66 depletion impairs the geometry of the cleavage furrow. U2OS cells were transfected with the indicated siRNAs, fixed with methanol, and stained for alpha-tubulin and DAPI. Images represent telophase and cytokinetic cells. Scale bar: 5 μm. (E) Quantification of (D). Asymmetric cleavage furrow indicates the cells as represented in (D) for siCCDC66. Skewed cells with asymmetric cleavage furrow are counted, and percentage is calculated according to total telophase/cytokinesis cell number. Data represent the mean ± SEM of 3

independent experiments. (\*\*$p < 0.01$). (F) In vitro microtubule bundling with full-length CCDC66. His-MBP-mNeonGreen-CCDC66 was purified from insect cells using Ni-NTA agarose beads, and microtubule bundling was performed. Alpha-tubulin was visualized by rhodamine-labeled tubulin, and CCDC66 was visualized with mNeonGreen signal. Scale bar: 5 μm. (G) In vitro microtubule bundling with CCDC66 (570–948). His-MBG-CCDC66 (570–948) C terminal fragment was purified from bacterial culture using Ni-NTA agarose beads, and microtubule bundling was performed. Alpha-tubulin was visualized by rhodamine-labeled tubulin. Scale bar: 5 μm. (H) Spatial distribution of PRC1 at the spindle midzone upon CCDC66 depletion. U2OS cells were transfected with the indicated siRNAs, fixed with methanol, and stained for alpha-tubulin, PRC1, and DAPI. Images represent anaphase cells, which are the colored versions of the inverted images presented in (A). Scale bar: 5 μm. The graph represents the plot profile of PRC1 and alpha-tubulin. Using ImageJ, a straight line (thickness 200) was drawn pole-to-pole direction covering the PRC1 area, and intensity along the distance was plotted on Graphpad Prism. Model shows representation of how the plot profile was generated. The data underlying the graphs shown in the figure can be found in S1 Data. CCDC66, coiled-coil domain-containing protein 66; DAPI, 4′,6-diamidino-2-phenylindole; MBP, maltose-binding protein; PRC1, protein regulator of cytokinesis 1; SEM, standard error of mean; siRNA, small interfering RNA.

direct MT affinity. Given that the C-terminal 570–948 residues of CCDC66 binds to MTs and localizes to spindle MTs, we next asked whether this fragment bundles MTs in vitro. We expressed and purified MBP-CCDC66 (570–948) in bacteria (S5G Fig). Like full-length CCDC66, MBP-CCDC66 (570–948) associated with MTs directly and promoted MT bundling in vitro (Figs 5G and S5H). In agreement with their in vitro activities, overexpression of mNG fusions of CCDC66 and its C-terminal (570–948) fragment induced formation of MT bundles (S5I Fig) in cells. Collectively, these results demonstrate that CCDC66 is a cross-linking MAP that is required for assembly and organization of central spindle and midbody MTs during cytokinesis.

MT-binding and bundling activities of PRC1 are required for microtubule organization at the spindle midzone in anaphase, localization of MAPs within this structure and successful completion of cytokinesis [33,67–69]. Given that CCDC66 interacts and co-localizes with PRC1 (Fig 2D and 2F), we tested whether the MT disorganization at the spindle midzone and cleavage furrow are due to defective PRC1 targeting to the central spindle and midbody. As revealed by the plot profile analysis of PRC1 intensity, the spatial distribution of PRC1 at the central spindle was disrupted. Specifically, PRC1 signal was spread over a broader region of anaphase B spindle in CCDC66-depleted cells (Fig 5H). Of note, the fluorescence intensity of PRC1 at the midbody was comparable between control and CCDC66-depleted cells (S5B Fig). Taken together, these results suggest potential involvement of CCDC66 in regulating recruitment of central spindle components including but not limited to PRC1.

## CCDC66 is required for centrosome maturation and MT nucleation during cell division

One of the most prominent phenotypes associated with CCDC66 depletion is the significant decrease in astral, metaphase spindle, and central spindle MT intensities. This finding led us to investigate the functions of CCDC66 during MT nucleation in dividing cells. To this end, we performed MT regrowth experiments in control and CCDC66-depleted cells synchronized using the Eg5-inhibitor S-trityl-L-cysteine (STLC) (Fig 6A). Following MT depolymerization by nocodazole treatment and its washout, cells were then fixed and stained for MTs and the centriole marker Centrin 3 at the indicated time points (Fig 6A). The centrosomal and non-centrosomal MT aster size was reduced in CCDC66-depleted cells relative to control cells at 3, 5, and 8 min after washout, which indicates MT nucleation defects (Fig 6A and 6B). Notably, CCDC66-depleted cells had an increased number of MT nucleating centers than control cells, suggesting possible activation of noncentrosomal MT nucleation pathways (S6A Fig). CCDC66-depleted cells were also delayed in the formation of the bipolar spindle after nocodazole treatment (Fig 6C and 6D). By 40 min, 30.9 ± 2.6% control cells and 9.9 ± 0.2%

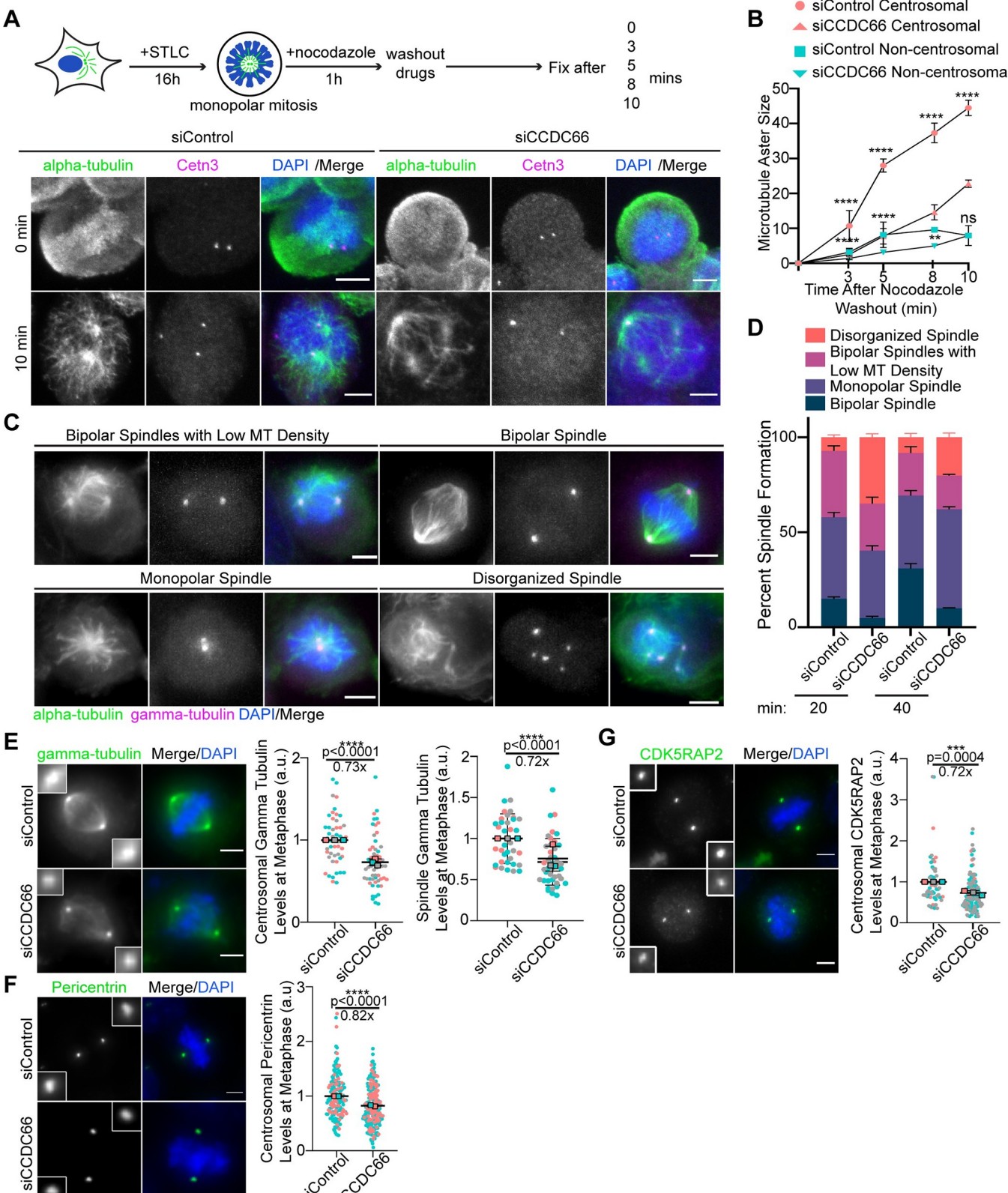

**Fig 6. CCDC66 recruits core PCM proteins to the centrosome and is required for mitotic microtubule nucleation.** (A) CCDC66 depletion slows down microtubule nucleation in mitotic cells. As illustrated in the experimental plan, U2OS cells were transfected with control and CCDC66 siRNA. Approximately

48 h post-transfection, they were synchronized with 5 μm STLC treatment for 16 h. After cell synchronization, microtubules were depolymerized by nocodazole treatment for 1 h. Following nocodazole wash out, cells were fixed and stained for alpha-tubulin, Centrin 3, and DAPI at the indicated time points. (B) Quantification of (A). For quantification of microtubule aster size, centrosomal and acentrosomal microtubule nucleation area is measured on ImageJ using polygon selection tool. Data represent the mean ± SEM of 2 independent experiments. (****$p < 0.0001$, **$p < 0.01$). Scale bar: 5 μm. (C) Effect of CCDC66 depletion on bipolar spindle assembly. Control and CCDC66 siRNA-transfected cells were stained with alpha-tubulin, gamma-tubulin, and DAPI. (D) Quantification of (C). Mitotic cells were scored based on their spindle architecture as bipolar spindle, monopolar spindle, bipolar spindles with low microtubule density, and disorganized spindle. Data represent the mean ± SEM of 2 independent experiments. Scale bar: 5 μm. (E–G) Effects of CCDC66 depletion on abundance of PCM proteins at the centrosomes. U2OS cells were transfected with control and CCDC66 siRNA. After 48 h, cells were fixed with methanol and stained for (C) gamma-tubulin, (D) CDK5RAP2, (E) pericentrin, and DAPI. Centrosomal abundance of PCM proteins and spindle abundance of gamma-tubulin were measured on ImageJ by drawing a 3.4 μm$^2$ circular area. Data represent mean ± SEM of 2 (pericentrin) and 3 (gamma-tubulin, CDK5RAP2) independent experiments. (****$p < 0.0001$). Images for each panel represent cells captured with the same camera settings from the same coverslip. Scale bar: 5 μm. The data underlying the graphs shown in the figure can be found in S1 Data. CCDC66, coiled-coil domain-containing protein 66; CDK5RAP2, CDK5 regulatory subunit-associated protein 2; DAPI, 4′,6-diamidino-2-phenylindole; SEM, standard error of mean; siRNA, small interfering RNA; STLC, S-trityl-L-cysteine.

CCDC66-depleted cells formed a bipolar spindle (Fig 6C and 6D). Supporting its roles in centrosome-mediated and noncentrosomal MT nucleation, gamma-tubulin levels at the centrosomes and microtubules were reduced in CCDC66-depleted cells relative to control cells (Fig 6E). Notably, ultrastructure expansion microscopy (U-ExM) analysis of STLC-synchronized G2 and mitotic cells confirmed reduced gamma-tubulin recruitment to the PCM and spindle MTs upon CCDC66 depletion (S6C Fig). We also noted that organization of gamma-tubulin at the PCM was disrupted while its pool at the centriole wall and lumen remained intact (S6C Fig), which was recently reported to be required for centriole integrity and cilium assembly [70].

During centrosome maturation, centrosomes recruit more PCM proteins to increase their MT-nucleating capacity. CDK5RAP2, CEP192, CEP152, and pericentrin are required for centrosomal recruitment of gamma-tubulin [13,71]. Given that CCDC66 interacts with these proteins, we tested their centrosomal targeting as a potential mechanism by which CCDC66 regulates centrosome maturation. To test this, we quantified centrosomal levels of these PCM proteins in control and CCDC66 siRNA-transfected cells. The levels of CDK5RAP2 and pericentrin, but not CEP192 and CEP152, were significantly reduced at the centrosomes in CCDC66-depleted cells as compared to control cells (Figs 6F, 6G, S6D, and S6E). Immunoblot analysis of lysates from control and CCDC66-depleted cells with antibodies against these proteins indicated that CCDC66 loss does not alter their cellular abundance (S6B Fig). Collectively, these results demonstrate that CCDC66 functions during centrosomal and noncentrosomal MT nucleation via targeting gamma-tubulin to centrosomes and microtubules.

## Expression of CCDC66 and its centrosome and MT-binding fragments restore mitotic and cytokinetic defects in CCDC66-depleted cells to different extents

The functional significance of the dynamic localization of CCDC66 to the centrosomes and various MT arrays as well as the relative contribution of its MT nucleation and organization activities to its functions are not known. To distinguish between the function of these different CCDC66 pools and activities during cell division, we performed phenotypic rescue experiments with 3 different CCDC66 siRNA-resistant constructs: (1) mNG-CCDC66 to validate the specificity of the phenotypes; (2) mNG-CCDC66 (570–948) to assess the functional significance of CCDC66 localization to the centrosomes and MT-binding and bundling activity; and (3) C-terminal mNG-CCDC66 fusion with the centrosomal targeting domain PACT to distinguish its centrosome-specific activities from the ones mediated by MTs. Next, we used lentiviral transduction to generate U2OS cells stably expressing these fusion proteins as well as

mNG itself as a control and validated their expression at the expected size, siRNA resistance, and localization in CCDC66-depleted cells by immunoblotting and immunofluorescence (Figs 7A, S7A, and S7B). In control and CCDC66 siRNA-transfected cells, mitotic localization profiles of mNG-CCDC66 and mNG-CCDC66 (570–948) were similar to the ones we reported in Fig 1 (Fig 7A). As for mNG-CCDC66-PACT, its localization was restricted to the centrosome and did not localize to the MTs (Fig 7A). Of note, its centrosomal to cytoplasmic relative fluorescent intensity was much higher, indicating that the majority of cellular CCDC66 was sequestered at the centrosome (Fig 7A).

Next, we examined whether expression of mNG-fusions of CCDC66, CCDC66 (570–948), and CCDC66-PACT restores defective targeting of gamma-tubulin to the centrosomes, reduced spindle MT intensity in metaphase and anaphase cells, increased cold-sensitivity of K-fibers, spindle misorientation, disorganized central spindle, and increased binucleation in CCDC66-depleted cells. mNG-CCDC66 expression rescued all 7 phenotypes to comparable or greater levels to control siRNA-transfected mNG-expressing cells, indicating that these phenotypes are specific to CCDC66 depletion (Figs 7A–7F and S7C–S7F). Similarly, mNG-CCDC66 (570–948) expression partially or fully rescued all 7 phenotypes, suggesting that centrosomal and MT affinity is sufficient for CCDC66 functions in these processes (Fig 7A–7F). As for the spindle MT levels, not only CCDC66 (570–948) but also full-length CCDC66 resulted in higher averages relative to control cells, suggesting that their expression might promote these phenotypes via increased MT nucleation. Strikingly, mNG-CCDC66-PACT only restored gamma-tubulin levels at the centrosomes, spindle MT levels, and spindle orientation defects to comparable or higher levels than that of control cells (Fig 7A–7C). However, it did not rescue defects in K-fiber stability, central spindle MT intensities, and organization and cytokinesis, suggesting that the MT-binding and bundling activities of CCDC66 is required for CCDC66 functions at the K-fibers, central spindle, and cleavage furrow (Figs 7D–7F and S7D–S7F). Collectively, these results show that CCDC66 functions during mitosis and cytokinesis via regulating centrosomal and noncentrosomal MT nucleation as well as MT organization. Importantly, the relative contribution of these activities to different CCDC66 functions varies based on the mechanisms by which different MT arrays are assembled and organized.

## Discussion

In this study, we identified the centrosomal and ciliary MAP, CCDC66, as a new player of the machinery governing the assembly and organization of the mitotic and cytokinetic MT arrays and thereby cell cycle progression. As summarized in the model shown in Fig 8, our findings reveal 2 important roles of CCDC66 during cell division. First, CCDC66 is required for mitotic progression via regulation of spindle assembly, organization and orientation, levels of spindle MTs, K-fiber integrity, and chromosome alignment. Second, CCDC66 functions during cytokinesis in part by regulating assembly and organization of central spindle and midbody MTs. Our work provides new insights into the spatiotemporal regulation of the mitotic and cytokinetic events governed by the dynamic changes of the MT cytoskeleton and centrosomes.

The results of our study revealed MT nucleation and organization as the 2 major mechanisms by which CCDC66 functions during cell division. In mitotic cells, CCDC66 regulates centrosome maturation via recruitment of core PCM proteins and is required for MT nucleation from the centrosomes during bipolar spindle assembly and orientation. The inability of centrosome-restricted mNG-CCDC66-PACT to rescue central spindle, cytokinesis, and K-fiber defects show that CCDC66 does not regulate these processes via centrosomal MT nucleation. Furthermore, 2 lines of evidence indicate that CCDC66 is also involved in MT-mediated MT nucleation during spindle assembly. First, depletion of CCDC66 resulted in

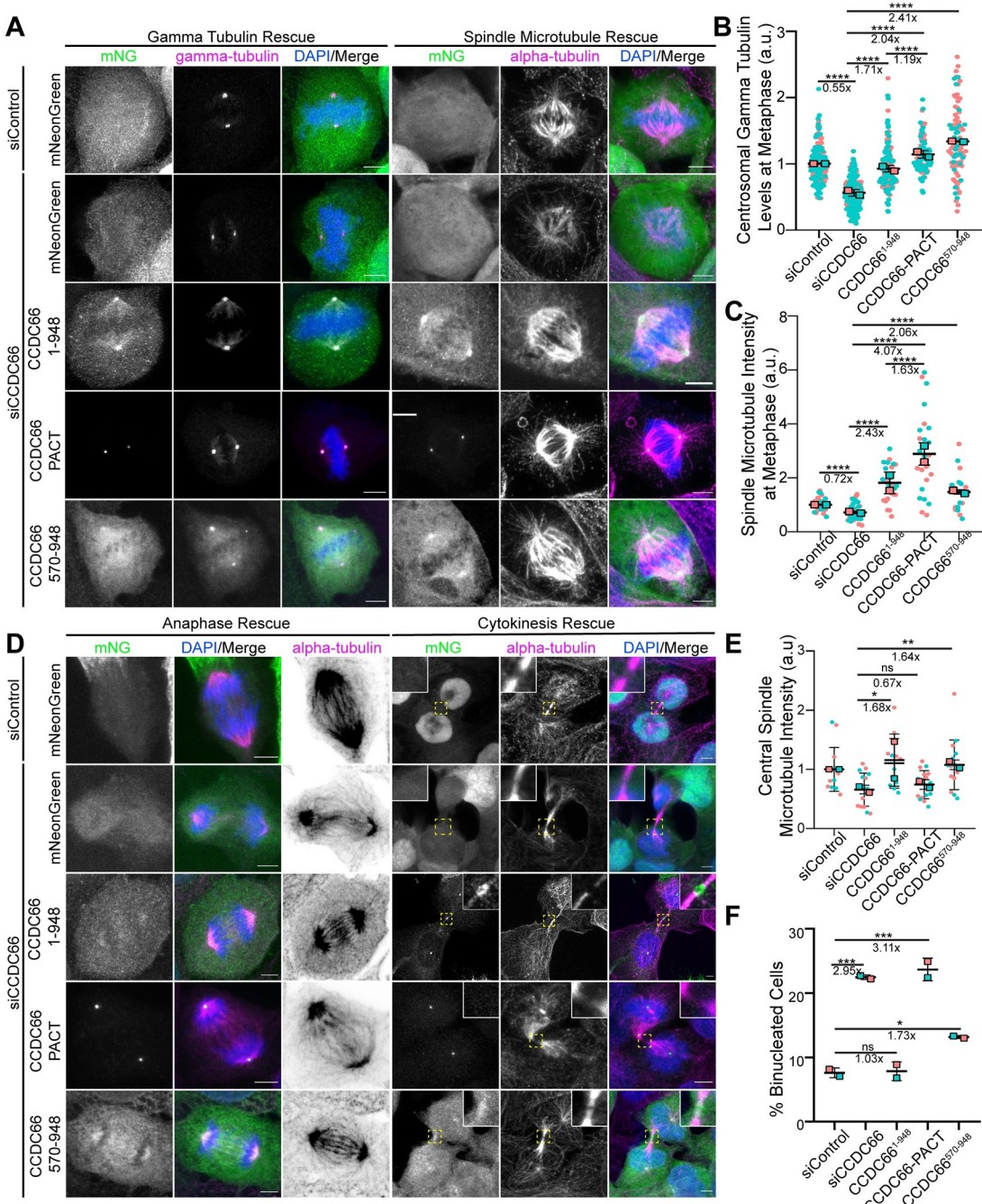

**Fig 7. Centrosome and microtubule affinity of CCDC66 is required for its mitotic and cytokinetic functions to different extents.** (A) Representative images for the gamma-tubulin and spindle microtubule density rescue experiments performed using U2OS::mNeonGreen, U2OS::mNeonGreen-CCDC66[1-948] (full-length), U2OS::mNeonGreen-CCDC66-PACT, and U2OS:: mNeonGreen-CCDC66[570-948] stable cells. Cells were transfected with control and CCDC66 siRNA. Approximately 48 h post-transfection, they were fixed with 4% PFA and stained for either gamma-tubulin or alpha-tubulin and DAPI. Scale bar: 5 μm. (B) Quantification of (A). Gamma-tubulin intensity was measured on ImageJ by drawing a 3.4 μm$^2$ circular area. Data represent the mean ± SEM of 2 independent experiments. (****$p < 0.0001$). (C) Quantification of (A). Spindle microtubule intensity was measured on ImageJ by taking several points on the spindle to measure the intensity then taking the average. Data represent the mean ± SEM of 2 independent experiments. (****$p < 0.0001$). (D) Representative images for the anaphase and cytokinesis rescue experiments performed using U2OS::mNeonGreen, U2OS::mNeonGreen-CCDC66[1-948], U2OS::mNeonGreen-CCDC66-PACT, and U2OS::mNeonGreen-CCDC66[570-948] stable cells. Cells were transfected with control and CCDC66 siRNA. Approximately 48 h post-transfection, they were fixed with 4% PFA and stained for alpha-tubulin and DAPI. Scale bar: 5 μm. (E) Quantification of (D). Graph represents the microtubule density at central spindle. Central spindle microtubule intensity was measured on

ImageJ by taking several points on the spindle to measure the intensity and subtracting the background mean intensity. Data represent the mean ± SEM of 2 independent experiments. (ns: not significant, $^*p < 0.05$, $^{**}p < 0.01$). (F) Quantification of (D). Graph represents the percentage of binucleated cell number. Data represent the mean ± SEM of 2 independent experiments. (ns: not significant, $^*p < 0.05$, $^{***}p < 0.005$). The data underlying the graphs shown in the figure can be found in S1 Data. CCDC66, coiled-coil domain-containing protein 66; DAPI, 4′,6-diamidino-2-phenylindole; PFA, paraformaldehyde; SEM, standard error of mean; siRNA, small interfering RNA.

a decrease in microtubule nucleation from noncentrosomal nucleation centers in mitotic cells as well as in gamma-tubulin recruitment to the spindle. Second, CCDC66 depletion resulted in decreased MT density of central spindle in anaphase, which represents de novo

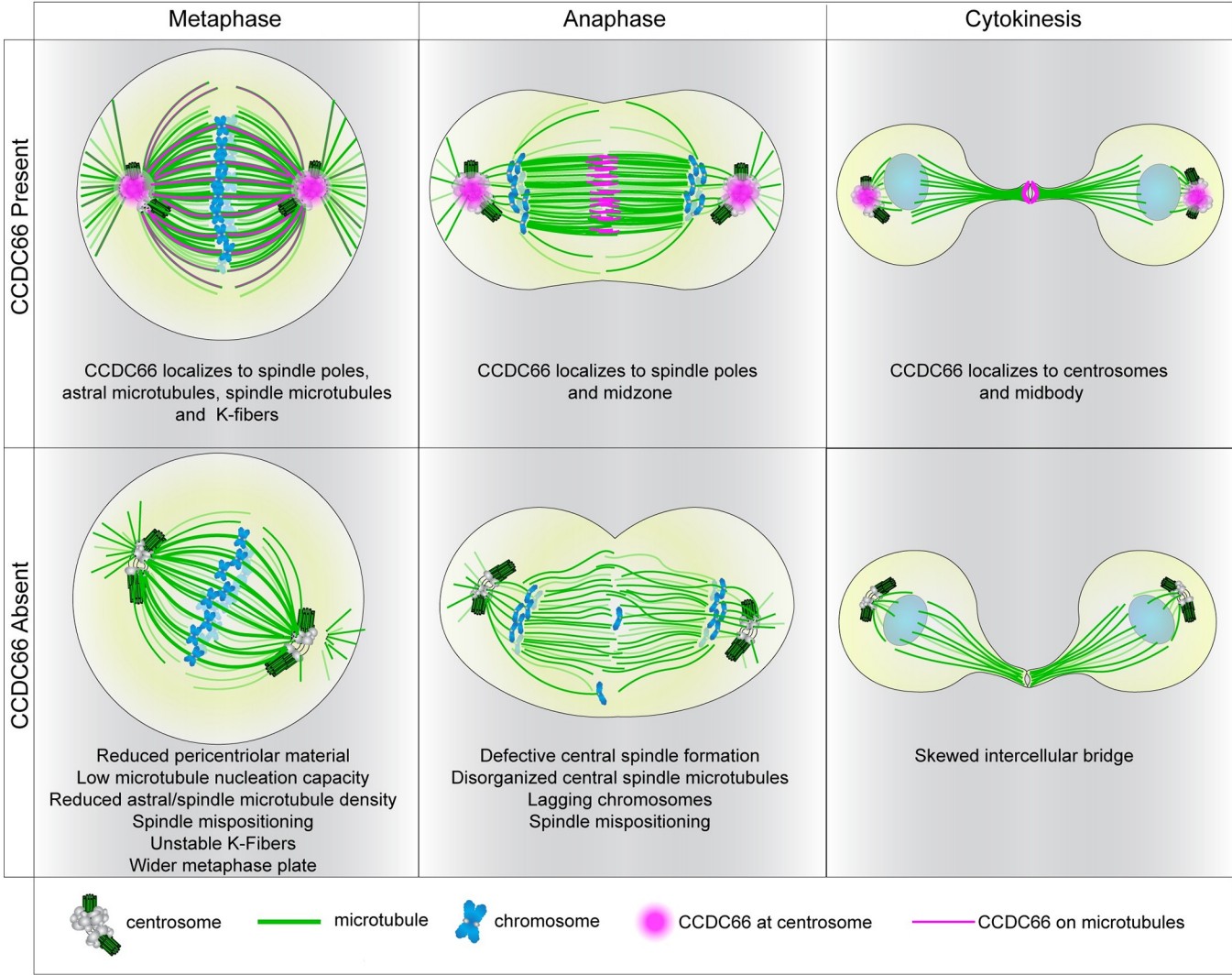

**Fig 8. Model CCDC66 localization and functions during mitosis and cytokinesis.** The model shows CCDC66 localization during cell division and the phenotypic consequences of CCDC66 depletion on mitotic and cytokinetic MT arrays as well as chromosomes. In metaphase cells, CCDC66 (magenta) localizes to the centrosomes, the astral microtubules, the spindle microtubules, and K-fibers. CCDC66 loss results in a reduced PCM size, microtubule nucleation capacity, astral/spindle microtubule density as well as unstable K-fibers. It also causes a shift in orientation and increases metaphase plate width. During anaphase, CCDC66 localizes to the centrosomes and midzone. CCDC66 depletion causes defective central spindle formation, reduction in central spindle microtubule intensity, lagging chromosomes, and orientation shift. During cytokinesis, CCDC66 additionally localizes to the centrosomes and the midbody. CCDC66 depletion causes asymmetric cleavage furrow formation. CCDC66, coiled-coil domain-containing protein 66; PCM, pericentriolar material.

noncentrosomal MT generation in inter-chromosomal region and requires MT-associated proteins that participate in nucleation, stabilization, and bundling [72,73]. The role of CCDC66 in MT-mediated MT nucleation is further supported by its proximity interactions with all 8 subunits of the HAUS/Augmin complex, which recruits γ-TuRC to MTs and promotes nucleation from spindle MTs [55]. Future studies are required to investigate whether and if so how CCDC66 regulates HAUS complex-dependent MT nucleation. It will be worthwhile to characterize direct contribution of CCDC66 on gamma and alpha/beta-tubulin recruitment and de novo MT formation in vitro in future studies.

The second mechanism by which CCDC66 operates during mitosis is organization of MTs via its cross-linking activity. Using in vitro and cellular experiments, we showed that CCDC66 bundles MTs in part via its C-terminal MT-binding domain. In line with our results, a recent study published during the revision of our manuscript reported that GFP-CCDC66 purified from mammalian cells binds to MTs and bundles them in vitro [74]. Bundling and stabilization of MTs is essential for the assembly and maintenance of the MT arrays such as the K-fibers, central spindle, and midbody [8]. Consistent with its in vitro MT bundling activity, CCDC66 depletion resulted in disorganized MTs at the K-fibers, central spindle, and cleavage furrow, and these defects were partially rescued by expression of mNG-CCDC66 (570–948), but not rescued by mNGCCDC66-PACT. While these results identify CCDC66 as a new player of organization of stable MT bundles during cell division, the mechanisms by which CCDC66 bundles MTs and the nature of these bundles are not known. Notably, CCDC66 interacts and co-localizes with PRC1 at the central spindle and midbody and its depletion disrupts PRC1 distribution at the central spindle, suggesting that CCDC66 regulates cytokinesis in part via PRC1. Further studies aimed at addressing how CCDC66 works together with PRC1 and other components of the central spindle and midbody, the factors contributing to contractile actomyosin ring (AMR) formation as well as elucidating the relative contribution of the N-terminal region of CCDC66 to its functions will be critical in providing mechanistic insight into CCDC66 functions during cytokinesis.

Despite its roles in MT nucleation and organization, CCDC66 depletion did not result in shorter spindles, which can be explained by a compensatory mechanism activated in CCDC66-depleted cells. Spatiotemporal regulation of MT polymerization, depolymerization, and sliding is critical to spindle length maintenance, providing remarkable ability of metaphase spindles to correct transient fluctuations in morphology [63,75]. Other MAPs and molecular motors that regulate MT stability, dynamics, sliding, as well as regulators of chromatid cohesion and chromosome MT nucleation, probably compensate and correct MT perturbation in CCDC66 absence to maintain steady-state spindle length [76]. Further characterization of the functional relationship of CCDC66 with the known mitotic MAPs and in vitro MT reconstitution assays will contribute to better understanding of regulation of spindle properties by CCDC66.

The pleiotropic localization and activities of CCDC66 during cell division present challenges in specifically defining mechanisms that underlie its mitotic and cytokinetic functions. Are they regulated by centrosomal and microtubule-associated pools of CCDC66 or their coordinated activity? While we first aimed to address this via identification of CCDC66 fragments that contribute to a single localization and activity, we could not identify such regions. As an alternative, we performed phenotypic rescue experiments with CCDC66 (570–948), which bundles MTs and localizes to centrosomes and CCDC66-PACT, which exclusively localizes to the centrosome. Of note, the strong centrosome affinity of the PACT domain increased CCDC66 levels at the centrosome and created more binding sites for its interactors such as gamma-tubulin, which might have compensated for lack of MT association for a subset of CCDC66 functions. Defective spindle MT density, recruitment of gamma-tubulin to the

centrosomes, and spindle orientation were rescued both by CCDC66-PACT and CCDC66 (570–948), albeit to different extents. These results suggest that centrosomal and MT pools cooperate in these processes. Strikingly, K-fiber, central spindle, and cytokinesis defects were rescued only by CCDC66 (570–948). Given the essential role of chromosome and MT-dependent MT nucleation and bundled MT arrays of the central spindle and midbody during cytokinesis, these results suggest that CCDC66-mediated MT bundling and noncentrosomal MT nucleation are required for these cellular processes [73,77].

CCDC66 has been implicated in several developmental disorders including retinal degeneration and Joubert syndrome [49–53]. Consistent with its link to ciliopathies, we and others previously showed that retinal degeneration mutations disrupt its ciliary functions and interactions [48,51]. Importantly, our results suggest that disruption of its nonciliary functions of CCDC66 might also contribute to disease pathogenesis. For example, CCDC66 is required for proper spindle orientation, which is critical for specification of the site of the cleavage furrow and distribution of cell fate determinants to daughter cells during architecture and organization of tissues affected in ciliopathies [78,79]. We note that CCDC66$^{-/-}$ mice were embryonically viable and did not develop tumors, indicating that defective cell cycle defects linked to CCDC66 loss alone do not disrupt embryogenesis and is not tumorigenic [50].This might be due to the compensatory mechanisms activated upon chronic loss of CCDC66, which is supported by its evolutionary conservation profile. CCDC66 is not as highly conserved as the HAUS complex or the core PCM proteins that it interacts with, suggesting that it is not an essential conserved player of cell division, but instead vertebrate-specific regulatory protein required for regulating the fidelity of cell division. Future studies are required to determine whether nonciliary functions of CCDC66 contribute to developmental disorders.

## Materials and methods

### Plasmids

pDEST-GFP-CCDC66 and pDEST-GFP-CCDC66$^{RR}$ plasmids used for transfection and transformation experiments were previously described [48]. Full-length CCDC66, CCDC66 (570–948), and CCDC66-PACT were amplified by PCR and cloned into pCDH-EF1-mNeonGreen-T2A-Puro lentiviral expression plasmid and pcDNA5.1-FRT/TO-FLAG-miniTurbo mammalian expression plasmid. CCDC66 (570–948) was cloned into pDEST-His-MBP expression plasmid using Gateway cloning (Thermo Scientific). siRNA-resistant mNG-CCDC66 was amplified from siRNA-resistant GFP-CCDC66$^{RR}$ plasmid and cloned into pCDH-EF1-mNeonGreen-T2A-Puro plasmid. EB3-mNeonGreen-T2A-gamma-tubulin-tagRFP plasmid was a gift from Andrew Holland (Johns Hopkins University School of Medicine) [80]. Myc-BirA* fusions of CDK5RAP2, CEP192, and CEP152 were previously described [81]. PLK1 was amplified by PCR and cloned into pDEST-Myc-BirA* plasmid using Gateway cloning. GFP-CEP55 plasmid was a gift from Kerstin Kutsche (UKE, Hamburg). GFP-PRC1 plasmid was a gift from Xuebiao Yao (Morehouse School of Medicine). mNeonGreen and mNeonGreen-CCDC66 were amplified by PCR and coned into pDONR221 using Gateway recombination. Subsequent Gateway recombination reactions using pDEST-His-MBP (gift from David Waugh—Addgene plasmid # 11085) and pFast-Bac-DEST (gift from Tim Stearns; Gateway R1R2 destination cassette was cloned into pFastBac-HT-MBP-D using KpnI and XhoI) were performed to generate His-MBP-mNeonGreen and His-MBP-mNeonGreen-CCDC66 for expression in bacterial cells and insect cells, respectively.

### Cell culture and transfection

Human telomerase immortalized retinal pigment epithelium cells (hTERT-RPE1, ATCC) were cultured with Dulbecco's modified Eagle's Medium DMEM/F12 50/50 medium (Pan

Biotech) supplemented with 10% fetal bovine serum (FBS, Life Technologies) and1% penicillin-streptomycin (Gibco). Human embryonic kidney (HEK293T, ATCC) and osteosarcoma epithelial (U2OS, ATCC) cells were cultured with DMEM medium (Pan Biotech) supplemented with10% FBS and 1% penicillin-streptomycin. All cell lines were authenticated by Multiplex Cell Line Authentication (MCA) and were tested for mycoplasma by MycoAlert Mycoplasma Detection Kit (Lonza). U2OS cells were transfected with the plasmids using Lipofectamine 2000 and according to the manufacturer's instructions (Thermo Scientific). HEK293T cells were transfected with the plasmids using 1 mg/ml polyethylenimine, MW 25 kDa (PEI). For microtubule depolymerization experiments, cells were treated with 5 µg/ml nocodazole (Sigma-Aldrich) or vehicle (dimethyl sulfoxide) for 1 h at 37˚C. For cell synchronization experiments, 5 µm (+)-S-trityl-L-cysteine (STLC) (Alfa-Aesar) was used for 16 h at 37˚C.

### Lentivirus production and cell transduction

Lentivirus were generated using pcDH-mNG-CCDC66, pcDH-mNG-CCDC66 (570–948), pcDH-mNG-CCDC66-PACT, and pCDH-EF1-mNeonGreen-T2A-Puro, and pLVPT2-mCherry-H2B plasmids as transfer vectors. For infection, $1 \times 10^5$ U2OS cells were seeded on 6-well tissue culture plates the day before infection, which were infected with 1 ml of viral supernatant the following day. Approximately 48 h post-infection, cells were split and selected in the presence of 6 mg/ml puromycin for RPE1s and 4 mg/ml puromycin for U2OS cells for 4 to 6 days until all the control cells died. U2OS and RPE1 cells stably expressing mCherry-H2B and mNeonGreen fusions of full-length CCDC66, and its truncations were generated by infection of cells with lentivirus expressing the fusions.

### siRNA and rescue experiments

CCDC66 was depleted using a siRNA with the sequence 5′-CAGTGTAATCAGTTCACAAtt-3′. Silencer Select Negative Control No. 1 (Thermo Scientific) was used as a control [48]. siRNAs were transfected into U2OS cells with Lipofectamine RNAiMax according to the manufacturer's instructions (Thermo Scientific). For rescue experiments, U2OS cells stably expressing mNG or mNG–CCDC66, mNG-CCDC66 PACT, and mNG-CCDC66 570–948 were transfected with control and CCDC66 siRNAs by using Lipofectamine RNAiMax (Thermo Scientific). Approximately 48 h post-transfection with the siRNAs, cells were fixed and stained. Due to the heterogeneity of the expression of fusion proteins, stable cells in which fusion proteins are not overexpressed and localize properly were accounted for quantification of phenotypic defects.

### Protein expression and purification

Protein expression in Rosetta (BL21(DE3)) cells transformed with respective constructs was induced by addition of 0.5 mM IPTG at OD600 of 0.5 to 0.6 for 16 h at 18˚C. BEVS baculovirus expression system and protocol was used for expression of tagged full-length CCDC66 protein. Briefly, 100 ml of Hi5 cells ($1 \times 10^6$ cell/ml) were infected with P1 baculovirus produced in Sf9 cells, carrying His-MBP-mNeonGreen-CCDC66, at MOI of 1. Cells were collected 48 h after infection.

For protein purification, cells were lysed by sonication in lysis buffer (20 mM Hepes pH 7.0 (or pH 7.5 for full-length protein), 250 mM NaCl, 0.1% Tween20, 2 mg/ml lysozyme, 1 mM PMSF, 1 mM protease inhibitor cocktail, 5 mM BME, and 10 mM Imidazole) and clarified at 19,000 rpm for 1 h at 4˚C. His-tagged proteins were subsequently purified using Ni-NTA–agarose beads (Thermo Scientific). Proteins were eluted with elution buffer (20 mM Hepes (pH

7.0 or pH 7.5), 250 mM NaCl, 5 mM BME, and 250 mM imidazole). For subsequent microtubule assays, proteins were dialyzed against BRB80 buffer (80 mM PIPES (pH 6.8), 1 mM EGTA, 1 m M MgCl$_2$).

## In vitro MT bundling assay

Fluorescent MTs were polymerized at 2 mg/ml by incubating tubulin and rhodamine-labeled tubulin (Cytoskeleton) at 10:1 ratio in BRB80 with 1 mM DTT and 1 mM GTP for 5 min on ice, then preclearing by centrifuging for 10 min at 90,000 rpm at 2˚C in TLA100 rotor. Cleared tubulin mixture was polymerized at 37˚C by adding taxol and increasing concentration stepwise, with final concentration of 20 μm. MTs were pelleted over warm 40% glycerol BRB80 cushion at 70,000 rpm for 20 min. After washes with 0.5% Triton-X100, pellet was resuspended in 80% of the starting volume of warm BRB80 buffer with 1 mM DTT and 20 μm taxol.

Bundling assays were performed as previously described [82]. Briefly, 100 nM of protein His-MBP-CCDC66570-948, His-MBP-mNeonGreen, His-MBP-mNeonGreen-CCDC66, or MBP was mixed with 2 μm MTs and 20 μm taxol in buffer T (20 mM Tris (pH 8.0), 150 mM KCl, 2 mM MgCl$_2$, 1 mM DTT, protease inhibitors) for 30 min rocking at room temperature. The reaction mixtures were transferred into a flow chamber under an HCl-treated coverslip, and unstuck proteins were washed out with the excess of buffer T. Bundling of fluorescent MTs was observed with a Leica SP8 confocal microscope and 63× 1.4 NA oil objective (Leica Mycrosystems). Experiment was repeated 4 times.

## In vitro MT sedimentation assay

Purified bovine brain tubulin (Cytoskeleton) was precleared at 90,000 rpm for 5 min at 2˚C with TLA100 rotor. Cleared tubulin was polymerized at 1 mg/ml in the presence of 1 mM GTP and increasing concentrations of taxol to a final concentration of 20 μm. After incubation of microtubules at room temperature overnight, 500 nM of purified protein was mixed with 3 μm taxol-stabilized microtubules or BRB80 buffer with taxol and incubated for 30 min at room temperature. Samples were loaded onto 60% glycerol BRB80 cushions and centrifuged at 50,000 rpm for 30 min with TLA100 rotor at room temperature. Supernatants were collected, glycerol cushions removed by aspiration, and pellets solubilized in SDS sample buffer. Equivalent volumes of supernatant and pellet fractions were resolved by SDS-PAGE.

## Ultra-structure expansion microscopy (U-ExM)

U-ExM was performed as previously described [83]. Briefly, U2OS cells were transfected with siControl or siCCDC66. Approximately 48 h after transfection, cells were treated with 5 μm STLC (Alfa-Aesar) for 16 h. Coverslips were incubated in 1.4% formaldehyde/2% acrylamide (2× FA/AA) solution in 1× PBS for 5 h at 37˚C prior to gelation in Monomer Solution supplemented with TEMED and APS (final concentration of 0.5%) for 1 h at 37˚C. Denaturation was performed at 95˚C for 1 h and 30 min, and gels were stained with primary antibodies for 3 h at 37˚C. Gels were washed 3 × 10 min at RT with 1 × PBS with 0.1% Triton-X (PBST) prior to secondary antibody incubation for 2.30 h at 37˚C followed by 3 × 10 min washes in PBST at RT. Gels were expanded in 3 × 150 ml dH2O before imaging. The following reagents were used in U-ExM experiment: formaldehyde (FA, 36.5% to 38%, Sigma-Aldrich), acrylamide (AA, 40%, Sigma-Aldrich), N,N'-methylenbisacrylamide (BIS, 2%, Sigma-Aldrich), sodium acrylate (SA, 97% to 99%, 408220, Sigma-Aldrich), ammonium persulfate (APS, 17874, Thermo Scientific), tetramethylethylenediamine (TEMED, Thermo Scientific), and poly-D-Lysine (Gibco).

## Immunofluorescence, antibodies, and microscopy

Cells were grown on coverslips, washed twice with PBS, and fixed in either ice cold methanol at −20˚C for 10 min or 4% PFA in Cytoskeletal Buffer ((100 mM NaCl (Sigma-Aldrich), 300 mM sucrose (Sigma-Aldrich), 3 mM MgCl2 (Sigma-Aldrich), and 10 mM PIPES (Sigma-Aldrich)). For CCDC66 endogenous staining with the rabbit polyclonal antibody, cells were first fixed with methanol at −20˚C, then with 100% acetone for 1 min at room temperature. After rehydration in PBS, cells were blocked with 3% BSA (Capricorn Scientific) in PBS followed by incubation with primary antibodies in blocking solution for 1 h at room temperature. Cells were washed 3 times with PBS and incubated with secondary antibodies and DAPI (Thermo Scientific) at 1:2,000 for 45 min at room temperature. Following 3 washes with PBS, cells were mounted using Mowiol mounting medium containing N-propyl gallate (Sigma-Aldrich). Primary antibodies used for immunofluorescence were rabbit anti-CCDC66 (Bethyl, A303-339A), mouse anti gamma-tubulin (Sigma, clone GTU-88, T5326) at 1:1,000, rabbit anti GFP at 1:2,000 (custom made) [81], mouse anti alpha-tubulin (Sigma-Aldrich, DM1A) at 1:1,000, rabbit anti CEP152 (Bethyl, A302-480A) at 1:500, rabbit anti-CEP192 (Proteintech, 18832 1 AP) at 1:1,000, rabbit anti-phospho-Histone H3 at 1:1,000, rabbit anti-CDK5RAP2 (Proteintech, 20061 1 AP) at 1:1,000, rabbit anti-pericentrin (Abcam, ab4448) at 1:2,000, rabbit anti-CSPP1 (Proteintech, 11931 1 AP) at 1:1,000, rabbit anti-PRC1 (Proteintech, 15617 1 AP) at 1:1,000, rabbit anti-Cep55 (Proteintech, 23891 1 AP) at 1:1,000, rabbit anti-Kif23 (Proteintech, 28587 1 AP), rabbit anti-pAurora A/B/C (Cell Signaling Technology, CST #2914) at 1:1,000, and mouse anti-mNeonGreen (Chromotek, 32F6) at 1:500, rabbit anti-GFP was generated and used for immunofluorescence as previously described [81]. Secondary antibodies used for immunofluorescence experiments were AlexaFluor 488-, 568-, or 633-coupled (Life Technologies) and they were used at 1:2,000.

Time lapse live imaging was performed with Leica SP8 confocal microscope equipped with an incubation chamber. For cell cycle experiments, asynchronous cells were imaged at 37˚C with 5% CO2 with a frequency of 6 min per frame with 1.5-mm step size and 12-mm stack size in 512 × 512 pixel format at a specific position using HC PL FLUOTAR 20×/0.50 DRY objective. For centrosomal protein level quantifications, images were acquired with Leica DMi8 inverted fluorescent microscope with a stack size of 8 mm and step size of 0.3 mm in 1,024 × 1,024 format using HC PL APO CS2 63× 1.4 NA oil objective. Higher resolution images were taken by using HC PL APO CS2 63× 1.4 NA oil objective with Leica SP8 confocal microscope.

Quantitative immunofluorescence for CEP192, CEP152, CDK5RAP2, pericentrin, gamma-Tubulin, PRC1, and alpha-tubulin was performed by acquiring a z stack of control and depleted cells using identical gain and exposure settings. The centrosome region for each cell was defined by staining for a centrosomal marker including gamma-tubulin. The region of interest that encompassed the centrosome was defined as a circle 3.4-mm$^2$ area centered on the centrosome in each cell. Total pixel intensity of fluorescence within the region of interest was measured using ImageJ (National Institutes of Health, Bethesda, Maryland). Background subtraction was performed by quantifying fluorescence intensity of a region of equal dimensions in the area proximate to the centrosome. Statistical analysis was done by normalizing these values to their mean.

## Spindle angle, length, and pole width measurements

U2OS cells were grown on coverslips and transfected with control and CCDC66 siRNA, fixed with methanol, and stained for gamma-tubulin. Images are acquired with Leica DMi8 inverted fluorescence microscope at 1,024 × 1,024 format with 0.3-mm z step size. Spindle angle is

calculated by the formula $\alpha = 180^*\tan^{-1}(h/L)/\pi$ where h represents the z stack difference between 2 centrosomes, L represents the distance between centrosomes. Spindle length is calculated by the formula $SL = \sqrt{(h2+L2)}$, where h represents the stack difference between 2 centrosomes, L represents the distance between centrosomes. Centrosome width at metaphase is calculated by measuring the length of the PCM of the centrosomes as described in the illustrations in Fig 4.

## Quantitative analysis of the spindle and astral microtubule intensity

U2OS cells were grown on coverslips and transfected with control and CCDC66 siRNA, fixed with methanol, and stained for alpha-tubulin. Images were acquired with Leica SP8 Confocal microscopy at $1,024 \times 1,024$ format with 2× zoom factor. For quantification of the astral microtubule intensity, 5 ROIs having XY size 2 microns were positioned manually on the astral microtubules and intensity was recorded. Background of the same ROI was measured in cytoplasm and subtracted from the average signal intensity. For astral microtubule length, the longest microtubule was measured using the length tool on ImageJ. For quantification of spindle microtubule intensity, 10 ROIs having XY size 2 microns were positioned manually on the spindle microtubules and intensity was recorded. Background of the same ROI was measured in cytoplasm and subtracted from the average signal intensity. The values of control and CCDC66 siRNA-treated metaphase cells were plotted relative to mean intensity of control siRNA. For K-fiber stability assay, control and CCDC66-depleted cells were placed on ice (4˚C) for 10 min without previous treatment and cell synchronization with proteasome inhibitor MG132. Cells were washed extensively with cold PBS to prevent MT polymerization, and then fixed with ice cold methanol at −20˚C for 3 min. The tubulin fluorescence intensity of cold stable K-fibers was measured as described above for quantification of spindle microtubule intensity.

## Quantitative analysis of the central spindle microtubule intensity and morphology

U2OS cells were grown on coverslips and transfected with control and CCDC66 siRNA, fixed with methanol, and stained for alpha-tubulin. Images were acquired with Leica SP8 Confocal microscopy at $1,024 \times 1,024$ format with 2× zoom factor. For quantification of central spindle microtubule intensity, 10 ROIs having XY size 2 microns were positioned manually on the central spindle microtubules and intensity was recorded. Background of the same ROI was measured in cytoplasm and subtracted from the average signal intensity. Aberrant central spindles are judged from the disorganization of the microtubules.

## Microtubule regrowth assay

U2OS cells were grown on poly-L-lysine coated coverslips and treated with control or CCDC66 siRNA. After 48 h of transfection, cells were treated with 5 μm/ml STLC for 16 h. Next day, cells were treated with 5 μg/ml nocodazole for 1 at 37˚C. Cells were washed extensively with cold PBS to prevent MT polymerization then incubated with warm media and fixed at indicated time points and stained for alpha-tubulin and centrin. Images were acquired with Leica SP8 Confocal microscopy at $1,024 \times 1,024$ format with 2× zoom factor. To quantify MT nucleation area (aster size), polygon selection tool on ImageJ is used. Centrosomal and noncentrosomal MT nucleation points were defined based on centrin staining, which marks the centrioles.

## Immunoprecipitation

HEK293T cells were co-transfected with indicated plasmids. Approximately 48 post-transfection, cells were washed and lysed with lysis buffer (50 mM HEPES (pH 8), 100 mM KCl, 2 mM EDTA, 10% glycerol, 0.1% NP-40, Protease inhibitors 1:100 pro-block PIC + 1:100 PMSF) for 45 min. Lysates were centrifuged at 13,000 rpm for 10 min at 4˚C, and supernatants were transferred to a tube. A total of 100 μl from each sample was saved as input. The rest of the supernatant was immunoprecipitated with anti-FLAG M2 agarose beads (Sigma-Aldrich) overnight at 4˚C. After washing 3× with lysis buffer, samples were resuspended in SDS containing sample buffer and analyzed by immunoblotting.

## Cell lysis and immunoblotting

Cells were lysed in 50 mM Tris (pH 7.6), 150 mM NaCl, 1% Triton X-100, and protease inhibitors for 30 min at 4˚C followed by centrifugation at 15.000$g$ for 15 min. The protein concentration of the resulting supernatants was determined with the Bradford solution (Bio-Rad Laboratories, California, United States of America). For immunoblotting, equal quantities of cell extracts were resolved on SDS-PAGE gels, transferred onto nitrocellulose membranes, blocked with TBST in 5% milk for 1 h at room temperature. Blots were incubated with primary antibodies diluted in 5% BSA in TBST overnight at 4˚C, washed with TBST 3 times for 5 min and blotted with secondary antibodies for 1 h at room temperature. After washing blots with TBST 3 times for 5 min, they were visualized with the LI-COR Odyssey Infrared Imaging System and software at 169 mm (LI-COR Biosciences). Primary antibodies used for immunoblotting were mouse anti gamma-tubulin (Sigma, clone GTU-88, T5326) at 1:5,000, rabbit anti GFP at 1:10,000 (homemade), mouse anti alpha-tubulin (Sigma, DM1A) at 1:5,000, rabbit anti CEP152 (Bethyl, A302-480A) at 1:1,000, rabbit anti-CEP192 (Proteintech, 18832 1 AP) at 1:1,000, rabbit anti-Cdk5Rap2 (Proteintech, 20061 1 AP) at 1:1,000, rabbit anti-pericentrin (Abcam, ab4448) at 1:2,000, mouse anti-mNG (Chromotek, 32F6) and mouse anti-CCDC66 (sigma SAB1408484) at 1:500. Secondary antibodies used for western blotting experiments were IRDye680- and IRDye800-coupled and were used at 1:15,000 (LI-COR Biosciences). Secondary antibodies used for western blotting experiments were IRDye680- and IRDye800-coupled and were used at 1:15,000 (LI-COR Biosciences)

## Quantification and statistical analysis

Data were analyzed and plotted using GrapPad Prism 7 (GraphPad, La Jolla, CA). Results are shown as mean ± standard errors (SEM) of the mean. Numbers of biological replicates are indicated in the figure legends. Two-tailed unpaired $t$ tests and 1-way analysis of variance (ANOVA) were applied to compare the statistical significance of the measurements. For data that does not follow normal distribution, we applied nonparametric Mann–Whitney test. Error bars reflect SD. Following key is followed for asterisk place holders for $p$-values in the figures: $^*p < 0.05$, $^{**}p < 0.01$, $^{***}p < 0.001$, $^{****} p < 0.0001$, ns: not significant.

## Supporting information

**S1 Fig. Dynamic CCDC66 localization during cell cycle and validation of mNG-CCDC66-expressing stable cell lines.** (A) Localization of CCDC66 at different stages of the cell cycle. U2OS were fixed with methanol followed by acetone and stained for CCDC66, alpha-tubulin, and DAPI. Scale bar: 5 μm, insets show 4× magnifications of the boxed regions. (B) Validation of RPE1::mNeonGreen-CCDC66 expression with antibody. RPE1::mNeonGreen-CCDC66 stable cell line was fixed with methanol and stained with

CCDC66 antibody and alpha-tubulin. Insets show 4× zoom. Scale bar: 5 μm. (C) Validation of mNeonGreen-CCDC66 expression in U2OS::mNeonGreen-CCDC66 and RPE1::mNeon-Green-CCDC66 stable lines by immunoblotting. Extracts from cells were prepared, resolved by SDS-PAGE and blotted with mNeonGreen antibody. (D) Relative expression level of mNeonGreen-CCDC66 compared to endogenous protein in U2OS cells. Extracts from cells were prepared, resolved by SDS-PAGE and blotted with CCDC66 antibody. (E) Localization of mNeonGreen-CCDC66 at different stages of cell cycle. RPE1 cells stably expressing mNeon-Green-CCDC66 fusion (RPE1::mNeonGreen-CCDC66) were fixed with 4% PFA and stained for alpha-tubulin and DAPI. Scale bar: 5 μm. (F) Dynamic localization of mNeonGreen-CCDC66 throughout the cell cycle. U2OS cells stably expressing mNeonGreen-CCDC66 fusion (U2OS::mNG-CCDC66) were incubated with 100 nM SiR-Tubulin overnight. Images are acquired every 4 min using confocal microscopy. Shown are sixteen time-lapse images from S2 Movie at indicated time points to show dynamic localization of mNeonGreen-CCDC66 to spindle poles and microtubules during cell division. (G) Dynamic localization of mNG-CCDC66 throughout the cell cycle. RPE1::mNG-CCDC66 were incubated with 100 nM SiR-Tubulin overnight. Images were acquired every 2 min using confocal microscopy. Shown are 14 time-lapse images from S1 Movie at the indicated time points. CCDC66, coiled-coil domain-containing protein 66; SiR-tubulin, silicon rhodamine (SiR) tubulin.
(TIF)

**S2 Fig. CCDC66 localizes to the central spindle and has extensive proximity interactions with regulators of cells division.** (A) Localization of mNeonGreen-CCDC66 in RPE1 cells relative to PRC1 during anaphase. RPE1::mNeonGreen-CCDC66 cells were fixed with 4% PFA and stained for PRC1 and DAPI. Graphs show the plot profiles to assess co-localization with the indicated marker. Using ImageJ, a straight line was drawn on the midbody, and intensity along the distance was plotted on Graphpad Prism. (B, C) GO-enrichment analysis of the CCDC66 proximity interactors based on their (B) biological process and (C) cellular compartment. The x-axis represents the log-transformed $p$-value (Fisher's exact test) of GO terms. The data underlying the graphs showing the plot profiles in the figure can be found in S1 Data. CCDC66, coiled-coil domain-containing protein 66; PFA, paraformaldehyde; PRC1, protein regulator of cytokinesis 1; DAPI, 4′,6-diamidino-2-phenylindole.
(TIF)

**S3 Fig. Validation of efficient CCDC66 depletion and its phenotypic consequences on mitotic fate and index.** (A) Validation of the efficiency of RNAi-mediated CCDC66 depletion by western blotting and immunofluorescence. U2OS cells were transfected with siControl or siCCDC66. Approximately 48 h after post-transfection, cells were fixed and stained with the indicated antibodies. In parallel, cell extracts immunoblotted for CCDC66 and vinculin (loading control). Band intensities were measured on ImageJ and normalized against background and vinculin intensities. Arbitrary value is determined based on siControl. (B) Validation of the CCDC66 antibody by immunofluorescence. U2OS cells were transfected with control or CCDC66 siRNA, fixed with methanol followed by acetone 48 h post-transfection and stained for CCDC66 and alpha-tubulin. Representative images are shown at different stages of the cell cycle to indicate the decrease in the signal of CCDC66 upon siCCDC66 transfection. Insets show 4× zoom of boxed areas. Scale bar: 5 μm. (C) Quantification of Fig 3A. The fate of individual cells was plotted as vertical bars, where the height of the bar represents the mitotic time, and the color of the bars represent the different fates including successful division (gray), mitotic arrest (pink), and apoptosis (cyan); n > 200 cells from each condition was quantified per condition. (D) Effect of CCDC66 depletion on mitotic index. U2OS cells were transfected with control or CCDC66 siRNA, fixed with methanol 48 h post-transfection and stained for

the mitotic marker phospho-Histone3 (pH3), alpha-tubulin, and DAPI. Mitotic cells are counted based on DNA staining. Data represent the mean ± SEM of 2 independent experiments and n > 1,000 for all experiments. Mean mitotic cell number for siControl is 10.31 and mean mitotic cell number for siCCDC66 is 12.49. Representative images are shown. Scale bar: 5 μm. The data underlying the graphs shown in the figure can be found in S1 Data. CCDC66, coiled-coil domain-containing protein 66; siRNA, small interfering RNA; SEM, standard error of mean; DAPI, 4′,6-diamidino-2-phenylindole.
(TIF)

**S4 Fig. Effects of CCDC66 depletion on cellular abundance of tubulin and endogenous localization of CCDC66 to K-fibers.** (A) Cellular abundance of tubulin in control and CCDC66-depleted cells. Cells are transfected with either control or CCDC66 siRNA. After 48 h, the lysates are collected and immunoblotted for alpha-tubulin, CCDC66, and vinculin (loading control). (B) Quantification of (A). Data represent mean ± SEM of 2 independent experiments. (ns: not significant). (C) CCDC66 localizes on K-fibers. U2OS cells were grown on coverslips and incubated in ice for 10 min before fixation with methanol followed by acetone. Cells were stained for CCDC66 and ACA. Inset shows 4× zoom of boxed area. Scale bar: 5 μm. The data underlying the graphs shown in the figure can be found in S1 Data. CCDC66, coiled-coil domain-containing protein 66; siRNA, small interfering RNA; SEM, standard error of mean; K-fiber, kinetochore fiber.
(TIF)

**S5 Fig. Validation of CCDC66 purification, microtubule association, and bundling.** (A) Spindle midzone length is not altered by CCDC66 depletion. U2OS cells were transfected with control or CCDC66 siRNA, fixed with methanol followed 48 h post-transfection and stained for alpha-tubulin and DAPI. As shown in the representation, midzone length was measured as the distance between the most distant microtubule ends in the midzone. Data represent the mean ± SEM of 4 independent experiments. (ns: not significant). (B) CCDC66 depletion does not alter PRC1 midbody levels. U2OS cells were transfected with siRNA then fixed with methanol after 48 h and stained for PRC1, alpha-tubulin, and DAPI. Images represent cells from anaphase and cytokinesis. Scale bar: 5 μm. For quantification, PRC1 intensity was measured on ImageJ, the background signal was subtracted, and normalized value was multiplied with area. Arbitrary value was determined by normalizing against siControl. (ns: not significant). (C) His-MBP-mNeonGreen-CCDC66 purification. His-MBP-mNeonGreen-CCDC66 was purified from insect cells using Ni-NTA agarose beads. Coomassie staining shows the proteins in pellet, initial sample, flowthrough, wash, and elutions. (D) Validation of His-MBP-mNeon-Green-CCDC66 purification. Purified His-MBP-mNeonGreen-CCDC66 purification was run on SDS-PAGE and blotted with CCDC66 antibody. Arrow corresponds to the full-length His-MBP-mNeonGreen-CCDC66. (E) His-MBP-mNeonGreen-CCDC66 directly interacts with microtubules. His-MBP-mNeonGreen-CCDC66 was purified from insect cells, and in vitro microtubule pelleting was performed and visualized by Coomassie staining. BSA was used as negative control. S stands for supernatant, P stands for pellet. (F) Validation of MBP-mNeon-Green purification with Coomassie. (G) Validation of MBP-CCDC66 (570–948) purification with Coomassie. MBP-His-CCDC66 (570–948) was purified from bacterial culture using Ni-NTA agarose beads. Purified protein was run on SDS-PAGE. Coomassie staining and western blotting with anti-His antibody shows the purified protein. (H) MBP-CCDC66 (570–948) directly interacts with microtubules. MBP-His-CCDC66 (570–948) was purified from bacterial culture using Ni-NTA agarose beads and microtubule pelleting was performed. Coomassie staining is presented. S stands for supernatant, P stands for pellet. (I) Effect of mNeonGreen-CCDC66 and mNeonGreen-CCDC66 (570–948) overexpression on microtubule organization

and stability in cells. U2OS cells were transfected with mNeonGreen, GFP-CCDC66, or mNG-CCDC66 (570–948) expression plasmids. Approximately 24 h post-transfection, cells were treated with 5 μm nocodazole or 0.01% DMSO for 1 h, fixed with methanol, and stained for the indicated proteins and DNA. Scale bar: 5 μm. The data underlying the graphs shown in the figure can be found in S1 Data. BSA, bovine serum albumin; CCDC66, coiled-coil domain-containing protein 66; DMSO, dimethyl sulfoxide; MBP, maltose-binding protein; PRC1, protein regulator of cytokinesis 1; siRNA, small interfering RNA; SEM, standard error of mean; DAPI, 4′,6-diamidino-2-phenylindole.
(TIF)

**S6 Fig. CCDC66 depletion compromises centrosome maturation and microtubule nucleation.** (A) Quantification of microtubule nucleation sites following nocodazole washout of STLC-synchronized control and CCDC66-depleted cells. Representative images are shown for control and CCDC66-depleted cells. The graph indicates the number of microtubule nucleation points that are counted from Centrin 3 and alpha-tubulin signals. Data represent the mean ± SEM of 2 independent experiments. ($^{**}p < 0.01$) Scale bar: 5 μm. (B) Effect of CCDC66 depletion on the cellular abundance of PCM proteins. U2OS cells were transfected with siControl or siCCDC66, and 48 h after transfection extracts from cells were immunoblotted for CDK5RAP2, CEP192, CEP152, gamma-tubulin, pericentrin and vinculin (loading control), or GAPDH (loading control). Band intensities were measured on ImageJ and normalized against background and vinculin intensities. Data represent the mean ± SEM of 3 independent experiments. (C) U-ExM analysis of control and CCDC66 depleted cells. U2OS cells were transfected with control and CCDC66 siRNA. Approximately 48 h post-transfection, cells were synchronized by 16 h STLC treatment and prepared for imaging. Cells were stained for gamma-tubulin and acetylated tubulin, imaged using confocal microscopy and deconvolved. Mitotic and G2 cells were picked for representation. (D, E) Effects of CCDC66 depletion on abundance of PCM proteins at the centrosomes. U2OS cells were transfected with control and CCDC66 siRNA. After 48 h, cells were fixed with methanol and stained for (D) CEP192 and (E) CEP152. Centrosomal abundance of PCM proteins was measured as described in Fig 4D. Images for each panel represent cells captured with the same camera settings from the same coverslip. Data represent mean ± SEM of 2 (CEP152) or 3 (CEP192) independent experiments. (ns: not significant). Scale bar: 5 μm. The data underlying the graphs shown in the figure can be found in S1 Data. CDK5RAP2, CDK5 regulatory subunit-associated protein 2; CEP152, centrosomal protein of 152 kDa; CEP192, centrosomal protein of 192 kDa; CCDC66, coiled-coil domain-containing protein 66; GAPDH, glyceraldehyde 3-phosphate dehydrogenase; PCM, pericentriolar material; SEM, standard error of mean; siRNA, small interfering RNA; STLC, S-trityl-L-cysteine; U-ExM, ultrastructure expansion microscopy.
(TIF)

**S7 Fig. Phenotypic rescue for spindle orientation and K-fiber stability using stable lines expressing mNG-CCDC66 fusion constructs.** (A) Validation of U2OS cells lines that stably express siRNA-resistant mNeonGreen-CCDC66, mNeonGreen-CCDC66-PACT, and mNeonGreen-CCDC66 (570–948). Extracts from cells were prepared, resolved by SDS-PAGE, and blotted with mNeonGreen antibody. (B) Validation of siRNA resistance of the CCDC66 rescue constructs. U2OS cells were transfected with control and CCDC66 siRNA. Approximately 48 h post-transfection extracts from cells were prepared, resolved by SDS-PAGE, and blotted with CCDC66 antibody. The red arrowheads indicate endogenous CCDC66, which is masked due to higher expression of the fusion proteins and high background associated with the CCDC66 antibody. The green arrowheads indicate the mNeonGreen fusions of CCDC66. (C) Quantification of Fig 7A. Spindle angle was calculated by the formula α = 180*tan$^{-1}$(h/L)/

$\pi$ where h represents the stack difference between 2 centrosomes, L represents the distance between centrosomes when projected onto 1 z plane. Data represent the mean ± SEM of 2 independent experiments. ($^{**}p < 0.01$ $^{***}p < 0.001$ $^{****}p < 0.0001$). (D) Representative images for the K-fiber rescue experiment performed using U2OS::mNeonGreen, U2OS::mNeonGreen-CCDC66$^{1-948}$, U2OS::mNeonGreen-CCDC66-PACT, and U2OS::mNeon-Green-CCDC66$^{570-948}$ stable cells. Cells were transfected with control and CCDC66 siRNA. Approximately 48 h post-transfection, they were fixed with methanol and stained for alpha-tubulin and DAPI. Scale bar: 5 µm. (E) Quantification of Fig 7D. Graph represents the intensity of K-fibers with mean ± SEM of 2 independent experiments. ($^{****}p < 0.0001$, ns: not significant). (F) Quantification of Fig 7D. Graph represents the percentage of aberrant central spindle with mean ± SEM of 2 independent experiments. ($^{****}p < 0.0001$, ns: not significant). The data underlying the graphs shown in the figure can be found in S1 Data. CCDC66, coiled-coil domain-containing protein 66; K-fiber, kinetochore fiber; siRNA, small interfering RNA; SEM, standard error of mean.
(TIF)

**S1 Raw Images. Unprocessed images for blots shown in the paper.** Red boxes are used to show the represented blot in the figures. Red crosses show the parts of the blot that are not used in the figures.
(PDF)

**S1 Data. Numerical data for Figs 2D, 3B–3E, 3G–3I, 4A, 4C, 4D, 5B, 5C, 5E, 5H, 6B, 6D, 6E–6G, 7B, 7C, 7E, 7F, S2A–S2C, S3C, S3D, S4B, S5A, S5B, S6A, S6D, S6E, S7C, S7E, S7F.** Each tab in the file represents the corresponding figure panel produced based on the displayed data.
(XLSX)

**S1 Movie. RPE1::mNG-CCDC66 cells were incubated with 100 nm SiR-Tubulin overnight and imaged with Leica SP8 Confocal microscopy equipped with an incubation chamber.** Images are taken every 2 min. Scale bar: 5 µm.
(AVI)

**S2 Movie. U2OS::mNeonGreen-CCDC66 cells were incubated with 100 nm SiR-Tubulin overnight and imaged with Leica SP8 Confocal microscopy equipped with an incubation chamber.** Images are taken every 4 min. Scale bar: 5 µm.
(AVI)

**S3 Movie. mCherry-H2B::U2OS cells were transfected with nontargeting siRNA and imaged with Leica SP8 Confocal microscopy equipped with an incubation chamber with 20× objective.** Images were taken every 6 min for 16 h. Scale bar: 5 µm.
(AVI)

**S4 Movie. mCherry-H2B::U2OS cells were transfected with CCDC66 siRNA and imaged with Leica SP8 Confocal microscopy equipped with an incubation chamber with 20× objective.** Images were taken every 6 min for 16 h. The movie represents a CCDC66 depleted cell going through normal mitotic progression. Scale bar: 5 µm.
(AVI)

**S5 Movie. mCherry-H2B::U2OS cells were transfected with CCDC66 siRNA and imaged with Leica SP8 Confocal microscopy equipped with an incubation chamber with 20× objective.** Images were taken every 6 min for 16 h. The cell marked with an asterisk represents a

CCDC66 depleted cell going through apoptosis after delayed mitosis. Scale bar: 5 μm.
(AVI)

**S6 Movie. mCherry-H2B::U2OS cells were transfected with CCDC66 siRNA and imaged with Leica SP8 Confocal microscopy equipped with an incubation chamber with 20× objective.** Images were taken every 6 min for 16 h. This movie represents a CCDC66 depleted cell going through premature mitotic exit. Scale bar: 5 μm.
(AVI)

**S7 Movie. mCherry-H2B::U2OS cells were transfected with CCDC66 siRNA and imaged with Leica SP8 Confocal microscopy equipped with an incubation chamber with 20× objective.** Images were taken every 6 min for 16 h. This movie represents a CCDC66 depleted cell with prometaphase arrest and nonpersistent metaphase plate. Scale bar: 5 μm.
(AVI)

**S8 Movie. mCherry-H2B::U2OS cells were transfected with CCDC66 siRNA and imaged with Leica SP8 Confocal microscopy equipped with an incubation chamber with 20× objective.** Images were taken every 6 min for 16 h. This movie represents a CCDC66 depleted cell with a cytokinesis defect. Scale bar: 5 μm.
(AVI)

**S9 Movie. mCherry-H2B::U2OS cells were transfected with CCDC66 siRNA and imaged with Leica SP8 Confocal microscopy equipped with an incubation chamber with 20× objective.** Images were taken every 6 min for 16 h. This movie represents a CCDC66 depleted cell with a cytokinesis defect. Scale bar: 5 μm.
(AVI)

## Acknowledgments

We acknowledge the Firat-Karalar lab members for insightful discussions regarding this work and Dila Gulensoy and Ezgi Odabasi for cloning the mNG-CCDC66-PACT construct and Efe Begar for purifying mNG protein.

## Author Contributions

**Conceptualization:** Umut Batman, Jovana Deretic, Elif Nur Firat-Karalar.

**Data curation:** Umut Batman, Jovana Deretic, Elif Nur Firat-Karalar.

**Formal analysis:** Umut Batman, Jovana Deretic.

**Funding acquisition:** Jovana Deretic, Elif Nur Firat-Karalar.

**Investigation:** Umut Batman, Jovana Deretic, Elif Nur Firat-Karalar.

**Methodology:** Umut Batman, Jovana Deretic, Elif Nur Firat-Karalar.

**Project administration:** Elif Nur Firat-Karalar.

**Resources:** Umut Batman, Jovana Deretic, Elif Nur Firat-Karalar.

**Software:** Jovana Deretic.

**Supervision:** Jovana Deretic, Elif Nur Firat-Karalar.

**Validation:** Umut Batman, Jovana Deretic, Elif Nur Firat-Karalar.

**Visualization:** Umut Batman, Jovana Deretic, Elif Nur Firat-Karalar.

**Writing – original draft:** Umut Batman, Jovana Deretic, Elif Nur Firat-Karalar.

**Writing – review & editing:** Umut Batman, Jovana Deretic, Elif Nur Firat-Karalar.

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
