## [Editor Report · Decision Letter 0]

29 Oct 2021

Dear Dr Firat-Karalar, 

Thank you for submitting your manuscript entitled "CCDC66 controls mitotic progression and cytokinesis by promoting centrosome maturation and microtubule bundling" for consideration as a Research Article by PLOS Biology. Please accept my apologies for the delay in getting back to you as we consulted with an academic editor about your submission.

Your manuscript has now been evaluated by the PLOS Biology editorial staff, as well as by an academic editor with relevant expertise, and I am writing to let you know that we would like to send your submission out for external peer review.

Once your full submission is complete, your paper will undergo a series of checks in preparation for peer review. Once your manuscript has passed the checks it will be sent out for review. 

If your manuscript has been previously reviewed at another journal, PLOS Biology is willing to work with those reviews in order to avoid re-starting the process. Submission of the previous reviews is entirely optional and our ability to use them effectively will depend on the willingness of the previous journal to confirm the content of the reports and share the reviewer identities. Please note that we reserve the right to invite additional reviewers if we consider that additional/independent reviewers are needed, although we aim to avoid this as far as possible. In our experience, working with previous reviews does save time. 

If you would like to send your previous reviewer reports to us, please specify this in the cover letter, mentioning the name of the previous journal and the manuscript ID the study was given, and include a point-by-point response to reviewers that details how you have or plan to address the reviewers' concerns. Please contact me at the email that can be found below my signature if you have questions. 

Please re-submit your manuscript within two working days, i.e. by Oct 31 2021 11:59PM.

Kind regards,

Richard

Richard Hodge, PhD

Associate Editor, PLOS Biology

rhodge@plos.org

PLOS

---

## [Decision Letter · Decision Letter 1]

16 Dec 2021

Dear Dr Firat-Karalar,

Thank you very much for submitting your manuscript "The ciliopathy protein CCDC66 controls mitotic progression and cytokinesis by promoting centrosome maturation and microtubule bundling" for consideration as a Research Article at PLOS Biology. Please accept my sincere apologies for the delays that you have experienced during the peer review process. Your manuscript has been evaluated by the PLOS Biology editors, an Academic Editor with relevant expertise, and by four independent reviewers.

The reviews are attached below. You will see that the two previous reviewers from the Life Science Alliance note that the addition of the endogenous CCDC66 data has improved the manuscript, but ask that the previously suggested experiments are addressed according to the revision plan provided in the rebuttal. After discussions with the Academic Editor, we agree with Reviewer #3 that the proposed phenotypic rescue experiments are essential to consider a revision of the manuscript.

We also recruited two additional reviewers, as we previously mentioned. While they find the work potentially interesting, they raise concerns regarding several statements over-interpretating the results. They also think that additional mechanistic data is needed to directly assign a role for CCD66 during cytokinesis and microtubule nucleation.

In light of the reviews, we will not be able to accept the current version of the manuscript. We remain interested in your study and we would be willing to consider resubmission of a comprehensively revised version that thoroughly addresses all the reviewers' comments. We cannot make any decision about publication until we have seen the revised manuscript and your response to the reviewers' comments. Your revised manuscript would be sent for further evaluation by the reviewers.

We appreciate that these requests represent a great deal of extra work, and we are willing to relax our standard revision time to allow you six months to revise your manuscript. We expect to receive your revised manuscript within 6 months.

**IMPORTANT - SUBMITTING YOUR REVISION**

*Resubmission Checklist*

*Published Peer Review*

*PLOS Data Policy*

*Blot and Gel Data Policy*

Sincerely,

Richard

Richard Hodge, PhD

Associate Editor, PLOS Biology

rhodge@plos.org

PLOS

REVIEWS:

Reviewer #1: In this manuscript, the authors examine the contribution of the ciliopathy protein CCDC66 to mitosis and cytokinesis. The authors show that the protein localizes to mitotic microtubules in dividing cells including spindle kinetochore fibers, PCM and midbody using both immunofluorescence and expression of the tagged exogenous protein. CCDC66 contribution to the localization of other PCM components, binding partners and microtubule crosslinking activity of CCD66 are determined. For the most part the work provides new information that will be of interest to cell biologists. 

1. To study CCD66 in mitotic cells, the authors deplete the protein with siRNA, which is verified with an immunoblot and by immunofluorescence. To more clearly depict the mitotic fate of the cells that were depleted of CCDC66, it would be good to show each cell as a bar representing the duration of mitosis and indicating the fate. An example of this type of representation see Wong, et al Science 2015, Figure 4. 

2. The defects in astral microtubules and spindle positioning and the localization of CCDC66 at the centrosome are consistent with a role for CCDC66 in microtubule nucleation. This is not a surprising result for a protein at the PCM that binds to gamma tubulin. The images of astral microtubules in the stained cells are not particularly convincing, and better images would be nice to see. 

3. Given the ability of CCDC66 to bind and bundle microtubules and its co-localization with PRC1 in the midbody, the authors hypothesize that CCDC66 contributes to cytokinesis. Consistent with this, the authors state that a large percentage of cells depleted for CCDC66 display asymmetric furrows indicating a defect in cytokinesis. However, I had some questions about other aspects of cytokinesis. Midzone localization in anaphase cells is weak - it is present in the expressing cells but not apparent in the stained cells. This should be noted or clarified. CCDC66 interacts with PRC and co-localizes with PRC at the midbody in late telophase/cytokinesis. Do CCDC66 and PRC1 co-localize at the anaphase midzone? Given the staining in Fig 1, I think not. Please clarify. The additional "spots" of fluorescence of CCDC66 in the midzone (distal to the co-localization with PRC) should be quantified to be sure these are not aggregates of the expressed protein. What % of cells show these? Are they observed in fixed and stained cells or only the expressing cells? The data in Fig S3 show midzone microtubules, in control and CCDC66 depleted cells, but defects but not quantified. The examples suggest subtle differences given the cell-to-cell variation in midzone microtubule organization. This needs to be quantified or removed. The cytokinesis defects could result from defects in spindle formation that are manifest later on at cytokinesis. Another possibility is that CCDC66 contributes not to midzone organization in anaphase, but in the abscission process.

4. The authors use microtubule regrowth assays to ask if the loss of CCDC66 impacts aster size and spindle formation following washout. They use STLC treated cells presumably so that a mono-aster is measured. But in the figure, the CCDC66 siRNA cell shown has two spots of gamma tubulin - it did not form a monopolar spindle? Is this a common phenotype? is the quantification of microtubule length for single asters (presumably the centrosomes separate on washout)?

5. The authors use knockdown and rescue to determine if the full-length protein, the microtubule binding C-terminal fragment and/or CCDC66 that is targeted to the centrosome with a PACT domain can rescue three aspects of the depletion phenotype: gamma tubulin at poles, spindle microtubule intensity and spindle length. The conclusion of the rescue experiments is that all three constructs can rescue these phenotypes, suggesting that CCDC66 at the centrosome is sufficient. In the discussion the authors state that "full length and C-terminal domain rescued these phenotypes to a greater extent than the CCDC66PACT" - although that is not clear from figure 7. Spindle length looks very similar in all cases and the PACT domain construct results more spindle tubulin. Are the differences in gamma level, spindle length and microtubule intensity for the full-length and PACT constructs significantly different (from each other) not just as compared to the depleted (and not rescued) cells? The three phenotypes measured all relate to the microtubule nucleating activity; are cytokinetic defects seen with the PACT containing construct? Is microtubule binding contributing to microtubule nucleation? I suppose resolving these issues will require use of constructs that contribute to a single function - nucleation or microtubule binding -- and are thus likely outside the scope of the present study. Nevertheless, the authors should be as clear as possible about the results of these experiments, which are impacted by possible overexpression of the rescue construct and additional centrosome binding in the case of the PACT construct. 

Minor

I do not see CCD66 on astral microtubules in figure S1E; please provide single color image of the CCD66 fluorescence.

The overexpression of CCD66 clearly results in artefacts as characterized in Figure 1D; thus, it is important that the level of expression be shown for rescue experiments and that the authors draw conclusions based on endogenous protein localizations.

Page 9 the word "not" is missing when the interaction with mycBirA is described.

In Movie 4 there are two mitotic cells; the one to the left looks like it undergoes apoptosis and the one on the left appears to undergo unequal division. In movie 5 there are also two mitotic cells; presumably the cell on the right is not depleted? Please add additional information.

In figure 4 some panels have dots of two colors and others three; the colors represent different experimental trials—please clarify in the legend which experiments were repeated three (or two) times.

Reviewer #2 (previous LSA reviewer): The revised version of the manuscript by Batman et al contains improvements that justify now consideration for a journal of higher visibility (such as PLOS Biology). In particular, the new data on endogenous CCDC66 represent a significant step forward. 

However, several of my earlier suggestions were apparently not followed: 

- I had criticized that spindle length in projected images appears shorter if spindles are tilted, and suggested that the length should be re-calculated from 3D data sets. Despite an appeasing response in which the authors promised that they will re-calculate ("As suggested by the reviewer, we will reanalyze our raw images to take into account the impact of the tilt angle of the spindles on spindle length for Fig. 4"), the data in the new Figure 4 are identical to the old version of the manuscript. 

- Rescue experiments for the cytokinesis phenotype were suggested. The authors state "As suggested by the reviewer, we have started performing experiments…. We will quantify … in the revised manuscript." However, the revised manuscript does not contain any such experiments. 

- I had criticized that in Fig. 5D, establishing a category of "prometaphase-like spindles" is not precise enough to describe the phenotype. Now, the authors labelled the same figure with the description "prometaphase spindles". This terminology is equally unsatisfactory, since spindle formation can be a long procedure even during a regular prometaphase, starting from monopolar arrays (if centrosomes aren't already separated at the time of nuclear envelope breakdown), transiting to somewhat "disorganized" spindles, and finishing with bipolar spindles. The authors must tell in descriptive terms what they see, for example "spindles with unfocused poles", or "spindles with low MT density", or other. The present classification makes little sense, since a typical "prometaphase spindle" does not exist. 

Overall, I am not satisfied with the revisions, and I suggest that additional improvements should be made. 

Reviewer #3 (previous LSA reviewer): Thank you to the authors to review the manuscript. They have addressed a critical number of experiments to reassure some of the reviewers' concerns;

1-The authors have reassured the critical point of this work regarding the overexpression of CCD66. They show the endogenous level of CCDC66 by immunofluorescence and western blot in U2OS. Also, they demonstrate the localization to the endogenous levels at the spindle poles and microtubules. Moreover, they have examined the protein levels of endogenous versus exogenous in the U2OS stable cell line. 

2- The authors have included some essential controls in the manuscript, replaced some representative images, and edited the manuscript to clarify some conclusions.

The authors mention in the rebuttal letter the importance to clarify whether the microtubule bundling activity of the mNG-CCDC66 (570-948) is sufficient to rescue microtubule bundling associated mitotic phenotypes. I consider this phenotypic rescue experiment is essential, and I agree with the authors about the revision plan to discern this issue.

As I mentioned in my previous revision, the results presented in this work are attractive to the general cell biological community, especially the centrosome and cilia communities. The authors are supported by solid evidence of their work, and their proposal strategy to address reviewers' comments is well conducted.

Reviewer #4: In this manuscript the authors study the protein CCDC66 that was previously found to be localized on centrosome and MT and to be involved in mitotic spindle assembly (by the same team and others). Here the authors try to understand how CCDC66 participates to spindle organization. Although they found that CCDC66 is required to localize centrosome proteins important for spindle assembly (centrosome maturation) it remains puzzling to understand how it works. CCDC66 is also a MAP that bundles MT, and it is also as puzzling to reconcile these two functions. 

General comments

The manuscript is very confusing, the experiments are well done but often one does not understand their choice and the order in which they are done.

The results are often over-interpreted, sometimes even very badly interpreted.

The manuscript shows interesting results but too preliminary as it is.

CCDC66 seems to be involved in the nucleation of MTs from centrosomes. Without CCDC66 the bipolar spindle is formed, this spindle has a normal length but the global network of mitotic MTs is less dense. This is interesting but how does it work? There is no mechanism described that would explain this phenotype.

Also, the controls are not shown in the right places. Two examples (1) siRNA depletion should appear to control for the specificity of the localization observed with the anti-CCDC66 antibody (2) rescue should demonstrate the specificity of the observed phenotype.

However, the localization (fig1) is associated with a long discussion/interpretation on the possible function of CCDC66 while the control (siRNA) is only shown in figure S3.

Same for the depletion (fig 4) which is very much discussed and interpreted while the control rescue is only shown fig 7.

The manuscript must be rewritten in a more logical way and progression and additional experiment performed to propose a mechanism to explain the function of CCDC6. The manuscript also needs proofreading and correction of English

Specific comments

Page 8 the authors say 

"Like other PCM proteins, CCDC66 formed resolvable rings at the PCM (Fig. 2A, 2B)"

I do not see any ring of CCDC66 on the spindle poles. May be in prometaphase but the resolution is not good enough to conclude. 

Page 9

Is this sentence correct?

CCDC66 co-pelleted with myc-BirA* fusions of CDK5RAP2, Cep192, Cep152 and gamma-tubulin, but with the negative control myc-BirA* (Fig. 2E).

I think it should be 

but NOT with the negative control myc-BirA

Page 9 again

"FLAGminiTurbo did not co-pellet with myc-BirA* fusions of these positive interactions, and

CCDC66 also did not co-pellet with the MT plus-end-tracking protein EB1, confirming the

specificity of its interactions with PCM proteins (Fig. 2E)"

One cannot draw such conclusion: not co-pelleting with EB1 is not enough to confirm the specificity of the interaction with the PCM proteins, it might only "suggest"

Page 9 & 10

"Taken together, our results suggest that CCDC66 functions during mitosis and cytokinesis by regulating centrosome maturation, MT nucleation and/or organization"

This is an overinterpretation, I would rather write:

"Taken together, the localization of CCDC66 during mitosis and cytokinesis led us to test potential functions of CCDC66 in the regulation of centrosome maturation, MT nucleation and/or organization"

- Control

I would put the figure S3B in figure 1 as a control of the specificity of the antibodies used in fig1A (at least figure S3A and S3B in figure S1) this would reinforce the initial data.

- Page 11 Microtubule densities after CCDC66 siRNA

"In agreement, we noted that the MT arrays of the bipolar spindle, central spindle and midbody of CCDC66-depleted cells were disorganized and prominent defects in the assembly and organization of the bipolar spindle, central spindle and midbody such as disorganized MTs AND REDUCED MT INTENSITIES (Fig. 3E, S3D)."

positioning. Relative to control cells, CCDC66

"… depletion resulted in a minor decrease in average spindle lengths, which was measured

as pole-to-pole distance in metaphase cells (Fig. 4A)."

I disagree, I don't see any difference

Page 12

"The tubulin fluorescence intensity at the spindle decreased about 0.6-fold in CCDC66-depleted cells relative to control cells (Fig. 4B, 4C)."

This is pretty obvious and this is the phenotype the authors must concentrate on. 

- Page 12

Fig 4B it appears that the whole MT network is less dense, MT nucleation or stability is affected. It seems that CCDC66 siRNA affect all MT not specifically Astral MT of K fibers.

"This result identifies CCDC66 as a regulator of K-fiber integrity, which explains the chromosome alignment defects and mitotic failure observed in CCDC66-depleted cells (Fig. 3)."

This looks again like an overinterpretation

- Fig 5E, F and G

The images show a huge difference in the intensity of the centrosome labelling of gamma tub, pericentrin and CDK5RAP2 (IN PARTICULAR CDK5RAP2), whereas the data that represent mean ±SEM of two (Pericentrin) and three (gamma-tubulin, CDK5RAP2) independent experiments show very little difference. A comment might be necessary to say that the authors selected the best picture to illustrate the difference.

- Fig S5

"Notably, CCDC66-depleted cells had an increased number of MT "nucleating centers than control cells, suggesting possible activation of non-centrosomal MT nucleation pathways (Fig. S5A)"

This is a pretty good example demonstrating that numbers are useful, looking to the figure S5A one can find only one more nucleation point in cells depleted of CCDC66. I am not sure that it is a significant increase. However, the intensity of gamma tub labelling of each point increases in control cells but do not increase in CCDC66 depleted cells. This rather tells that nucleation of MT is affected. 

I would ask the authors to measure the intensity of gamma tub labelling of each point and compare with and without CCDC66.

- Fig 6 and page 14

"During the analysis of CCDC66-depleted cells stained for MTs, we noted that the cleavage furrow ingression is highly asymmetric and/or skewed in a significant number of cells (Fig. 6C)."

highly asymmetric and/or skewed?

The authors must be more precise

The telophase cleavage furrow shown in Fig6C bottom is it asymmetric, skewed, bent …? What is the % of such images …

Also it is difficult to draw any conclusion by only looking to and comparing the shape of the telophase cleavage furrow.

I invite the authors to read the following paper (Lafaurie-Janvore J, Maiuri P, Wang I, Pinot M, Manneville JB, Betz T, Balland M, Piel M. ESCRT-III assembly and cytokinetic abscission are induced by tension release in the intercellular bridge. Science. 2013 Mar 29;339(6127):1625-9)

- The rescue

These are very nice experiments, that strengthen the data

CCDC66-PACT rescue MT densities, this is surprising that the only localization at the centrosome is enough to rescue

570-948 also works but the protein is all over the place

I would like to see WB showing the level of expression of the different proteins (NG CCDC66 1-948) (NG CCDC66-PACT) & (NG CCDC66 570-948) in the rescue experiments.

« We performed rescue experiments for defective targeting of gamma-tubulin to the

spindle poles, reduced spindle tubulin intensity and SHORTER SPINDLE LENGTH associated with CCDC66 loss »

Well as I said in fig4A there is no detectable differences in the length of the spindle with or without CCDC66

---

## [Decision Letter · Decision Letter 2]

26 May 2022

Dear Dr Firat-Karalar,

Thank you for your patience while we considered your revised manuscript entitled "The ciliopathy protein CCDC66 controls mitotic progression and cytokinesis by promoting microtubule nucleation and organization" for publication as a Research Article at PLOS Biology. This revised version of your manuscript has been evaluated by the PLOS Biology editors, the Academic Editor and the previous four reviewers.

Based on the reviews, we are likely to accept this manuscript for publication, provided you satisfactorily address the remaining points raised by Reviewer 1. Please also make sure to address the following data and other policy-related requests stated below.

We expect to receive your revised manuscript within two weeks. 

*Published Peer Review History*

*Press*

Sincerely,

Ines

--

Ines Alvarez-Garcia, PhD

Senior Editor

PLOS Biology

on behalf of

Richard Hodge, PhD

Associate Editor,

rhodge@plos.org,

PLOS Biology

Fig. 2D; Fig. 3B-E, G-I; Fig. 4A, C, D; Fig. 5B, C, E, H; Fig. 6B, D-G; Fig. 7B, C, E, F; Fig. S2A, B; Fig. S3C, D; Fig. S4B; Fig. S5A, B; Fig. S6A, D, E and Fig. S7C, D, F

We require the original, uncropped and minimally adjusted images supporting all blot and gel results reported in an article's figures or Supporting Information files. We will require these files before a manuscript can be accepted so please prepare and upload them now. Please carefully read our guidelines for how to prepare and upload this data: https://journals.plos.org/plosbiology/s/figures#loc-blot-and-gel-reporting-requirements

Reviewers' comments

Rev. 1:

This revised manuscript has been substantially improved. The points raised in the initial review have been addressed.

Comments:

1. for the assay of kinetochore fiber stability, the authors used a brief cold treatment to disassemble the non-kinetochore associated microtubules. Cells at metaphase have k-fiber microtubules that resist disassembly; cells prior to metaphase do not. Did the authors include MG132 to arrest cells at metaphase, thus avoiding the possibility that prometaphase cells are included in the analysis?

2. what is the role of the N-terminal portion of CCDC66? The authors might state if there is a known role, as a reminder for the reader.

3. Mispositioning and angle. The authors measure spindle tilt, a feature that they noticed was altered in the siRNA depleted cells. In the text they use "mispositioning" in several places to refer to this phenotype. Some cells position their spindle to one side of the cell, but to my knowledge the cells used in this study have centrally positioned spindles and the position of the spindle relative to the cell cortex was not actually measured. I think the term mis/positioning should not be used unless the authors mean that the spindle is off-center in a cell with a centrally located spindle. The feature measured was the angle relative to the substrate.

4. Centrosome and spindle pole. I realize that these two terms are commonly used interchangeably, but they are not actually the same. The centrosome, consisting of the centrioles and PCM, localizes at the spindle pole for spindles in cells that have centrosomes.Spindle microtubules converge into the spindle pole and the components localized there are not centrosomal components. see: 10.1242/jcs.111.11.1477

On the top of page 9, the authors say that CCDC66 is at the centrosome in interphase and at the spindle pole and microtubule based structures in mitosis. In this instance, "at the centrosome" might be more accurate.

A bit later, they describe results using hi-resolution imaging to localize CCDC66 relative to markers of the "spindle pole" (top of page 10) and go on to say they used antibodies to PCM proteins. The first sentence in this section should read "centrosome" or mitotic centrosome, because that is what they are looking at. This section should be edited to refer to the centrosome.

The presentation of the data and the writing have improved the manuscript.

Rev. 2:

The manuscript by Batman et al characterizes the role of the protein CCDC66 in mitosis. The authors made the novel discovery of CCDC66 acting in centrosomal recruitment of nucleation factors, and in the bundling of microtubules at the midbody during cytokinesis. The manuscript should be of interest to a large community of cell biologists.

Compared to earlier versions of this submission, the authors made significant experimental improvements. All my earlier criticism has now been addressed and I have a favorable opinion on this manuscript.

Rev. 3:

I am happy that the authors have answered all of the points I raised during the first and second revisions.

I consider that the authors have revised the manuscript addressing the major and minor concerns from the two revisions. They have added knowledge about the ciliar and nonciliar CCDC66 functions at the centrosomes and microtubules during cell division.

Rev. 4:

The authors have greatly improved their manuscript. This revised version has a much more logical flow and is now very enjoyable to read. The data have also been improved in a very significant way by adding a new set of experiments and data, with new antibodies for example and quantifications. Questionable data have been removed or my questions addressed by new experiments. The rescue experiments strengthen the data regarding the function of CCD66 in spindle assembly through nucleation and microtubule organization.

This manuscript will certainly attract the attention of cell biologists working on microtubule spindle organization in mitosis.

Excellent work!

---

## [Editor Report · Decision Letter 3]

14 Jun 2022

Dear Elif,

On behalf of my colleagues and the Academic Editor, Renata Basto, I am pleased to say that we can accept your manuscript for publication, provided you address any remaining formatting and reporting issues. These will be detailed in an email you should receive within 2-3 business days from our colleagues in the journal operations team; no action is required from you until then. Please note that we will not be able to formally accept your manuscript and schedule it for publication until you have completed any requested changes.

PRESS

Thank you again for choosing PLOS Biology for publication and supporting Open Access publishing. We are looking forward to publishing your study. 

Best wishes, 

Richard

Richard Hodge, PhD

Associate Editor, PLOS Biology

rhodge@plos.org

PLOS
